# Placental cell type deconvolution reveals that cell proportions drive preeclampsia gene expression differences

Kyle A. Campbell [1], Justin A. Colacino [2,3], Muraly Puttabyatappa[4], John F. Dou[1], Elana R. Elkin [2], Saher S. Hammoud[5,6,7], Steven E. Domino[6], Dana C. Dolinoy[2,3], Jaclyn M. Goodrich [2], Rita Loch-Caruso[2], Vasantha Padmanabhan [2,3,4,6] & Kelly M. Bakulski [1✉]

The placenta mediates adverse pregnancy outcomes, including preeclampsia, which is characterized by gestational hypertension and proteinuria. Placental cell type heterogeneity in preeclampsia is not well-understood and limits mechanistic interpretation of bulk gene expression measures. We generated single-cell RNA-sequencing samples for integration with existing data to create the largest deconvolution reference of 19 fetal and 8 maternal cell types from placental villous tissue ($n = 9$ biological replicates) at term ($n = 40{,}494$ cells). We deconvoluted eight published microarray case–control studies of preeclampsia ($n = 173$ controls, 157 cases). Preeclampsia was associated with excess extravillous trophoblasts and fewer mesenchymal and Hofbauer cells. Adjustment for cellular composition reduced preeclampsia-associated differentially expressed genes ($\log_2$ fold-change cutoff $= 0.1$, FDR $< 0.05$) from 1154 to 0, whereas downregulation of mitochondrial biogenesis, aerobic respiration, and ribosome biogenesis were robust to cell type adjustment, suggesting direct changes to these pathways. Cellular composition mediated a substantial proportion of the association between preeclampsia and *FLT1* (37.8%, 95% CI [27.5%, 48.8%]), *LEP* (34.5%, 95% CI [26.0%, 44.9%]), and *ENG* (34.5%, 95% CI [25.0%, 45.3%]) overexpression. Our findings indicate substantial placental cellular heterogeneity in preeclampsia contributes to previously observed bulk gene expression differences. This deconvolution reference lays the groundwork for cellular heterogeneity-aware investigation into placental dysfunction and adverse birth outcomes.

[1] Epidemiology, School of Public Health, University of Michigan, Ann Arbor, MI, USA. [2] Environmental Health Sciences, School of Public Health, University of Michigan, Ann Arbor, MI, USA. [3] Nutritional Sciences, School of Public Health, University of Michigan, Ann Arbor, MI, USA. [4] Pediatrics, Michigan Medicine, University of Michigan, Ann Arbor, MI, USA. [5] Human Genetics, Michigan Medicine, University of Michigan, Ann Arbor, MI, USA. [6] Obstetrics and Gynecology, Michigan Medicine, University of Michigan, Ann Arbor, MI, USA. [7] Department of Urology, Michigan Medicine, University of Michigan, Ann Arbor, MI, USA. ✉email: bakulski@umich.edu

The public health burden of adverse pregnancy outcomes is substantial. An important example is preeclampsia, which affected 6.5% of all live births in the United States in 2017 and is characterized by high maternal blood pressure, proteinuria, and damage to other organ systems[1]. Adverse pregnancy outcomes may lead to myriad health complications including an elevated risk of chronic diseases throughout the life course[2]. The placenta, a temporary organ that develops early in pregnancy, promotes maternal uterine artery remodeling; mediates transport of oxygen, nutrients, and waste[3]; secretes hormones to regulate pregnancy; metabolizes various macromolecules and xenobiotics; and can serve as a selective barrier to some, but not all, pathogens and xenobiotics[4]. The executive summary of the Placental Origins of Adverse Pregnancy Outcomes: Potential Molecular Targets workshop recently concluded that most adverse pregnancy outcomes are rooted in placental dysfunction[5]. Despite this, the molecular underpinnings of placental dysfunction are poorly understood.

Placenta-specific cell types including cytotrophoblasts, syncytiotrophoblasts, extravillous trophoblasts, and placental resident macrophage Hofbauer cells are all essential for placental development, structure, and function[6]. Dysfunction of these specific cell types likely plays a role in placental pathogenesis. For example, extravillous trophoblasts are responsible for invading into the maternal decidua early in pregnancy to remodel uterine arteries and increase blood flow to the placenta[3]. Inadequate or inappropriate invasion of extravillous trophoblasts has previously been implicated in preeclampsia etiology[7–9]. Despite some knowledge of the roles of specific placental cell types in the development of preeclampsia, relatively little is known about how individual cell types contribute to placental dysfunction.

Existing research models used to investigate the function and dysfunction of individual cell types are limited. Protocols to isolate primary placental cells for experimental research are restricted to one or few cell types[10–15]. Cell type-specific assays are costly and require special techniques or training resulting in small sample sizes and have not yet been scalable to large epidemiological studies[16–18]. Furthermore, placental cell lines such as BeWo, derived from choriocarcinoma[19], and HTR-8/SVneo, immortalized by SV40[20], are typically derived by processes that alter the DNA of the cells, limiting their in vivo translatability. Consequently, the characteristics of even healthy placental cell type function and especially their connections to adverse outcomes such as preeclampsia are incompletely understood.

Measures of gene expression in bulk placental tissue are used to better understand the biological mechanisms underlying adverse pregnancy outcomes[21–23] and are common in epidemiological studies[24]. Gene expression profiles differ systematically by cell type[25,26]. Thus, bulk placental tissue-level gene expression measurements represent a convolution of gene expression signals from individual cells and cell types[27,28]. Deconvolution refers to the bioinformatic process of estimating the distribution of cell types that constitute the tissue[29,30]. Deconvoluting tissue-level gene expression profiles is essential to account for effects introduced by unmodeled cell type proportions[31] by disentangling shifts in cell type proportions from direct changes to cellular gene expression[32]. Reference-based deconvolution boasts biologically interpretable cell type proportion estimates with few modeling assumptions but relies on independently collected cell type-specific gene expression profiles as inputs[32]. Prior placental cell type-specific gene expression measures from term villous tissue[16,17] had a limited number of biological replicates and included neither technical replicates nor benchmarking against physically isolated placental cell types. A robust, accessible, and publicly available gene expression deconvolution reference is currently unavailable for healthy placental villous tissue.

To advance the field of perinatal molecular epidemiology, our goal was to develop an accessible and robust gene expression deconvolution reference for healthy placental villous tissue at term. We generated single-cell RNA-sequencing data with technical replicates for integration with existing cell type-specific placental gene expression data[16,17]. In addition, we benchmarked these single-cell cell type-specific gene expression profiles against placental cell types isolated with more conventional fluorescence-activated cell sorting (FACS) followed by bulk RNA-sequencing. Finally, to assess links between preeclampsia and placental cell types and their proportions, we applied our placenta cell type gene expression reference to deconvolute bulk placental tissues in a secondary data analysis of a case–control study[33] of preeclampsia, including a mediation analysis of the preeclampsia-associated genes *FLT1*, *LEP*, and *ENG* that quantifies the role cellular composition plays in explaining bulk gene expression measures.

## Results

### Single-cell gene expression map of healthy placental villous tissue.

A conceptual layout of the laboratory methods and analyses contained within this manuscript is provided in Supplementary Fig. 1. From healthy term placental villous tissue, 9244 cells across a total of two biological replicates and two technical replicates were sequenced and analyzed (Michigan sample). These data were combined with single-cell RNA-sequencing data of 5911 cells from three healthy term villous tissue samples in a previously published study (Pique-Regi sample)[17] and 25,339 cells from four healthy term villous tissue samples in another previously published study, two of which were subsampled with an additional peripheral placental villous tissue sample (Tsang sample)[16] (Supplementary Table 1). Cells were excluded if they had low RNA content (<500 unique RNA molecules), few genes detected (<200), or were doublets or outliers in mitochondrial gene expression (Supplementary Figs. 2 and 3). Fetal or maternal origin of cells was determined by genetic variation in sequencing data. Fetal sex was determined by *XIST* expression (Supplementary Fig. 4). The final analytic sample included 40,494 cells and 36,601 genes across nine biological replicates, two of which had a technical replicate and another two included peripheral subsampling.

Uniform Manifold Approximation and Projection (UMAP)[34] was used to visualize sequencing results in two dimensions with mutual nearest neighbor batch correction[35] (Fig. 1a). Cells clustered into 19 fetal and 8 maternal cell types with 84.4% of all cells being of fetal origin (Table 1). Cell type clustering decisions balanced cluster stability, resolution, and biologic plausibility with prior knowledge. If desired, downstream analyses could collapse cell subtypes into a single, more general cell type cluster. We observed placenta-specific trophoblast cell types including cytotrophoblasts (*KRT7*), proliferative cytotrophoblasts (*KRT7*, *STMN1* and other proliferation-related genes)[36], extravillous trophoblasts (*HLA-G*)[37], and syncytiotrophoblasts (*PSG4* and other pregnancy-specific hormone genes) (Supplementary Fig. 5a)[38]. Proliferative cytotrophoblasts were distinguished from other cytotrophoblasts by overexpression of genes related to cytoplasmic translation ($p_{adj} = 8.1 \times 10^{-15}$) and mitotic sister chromatin segregation ($p_{adj} = 1.5 \times 10^{-12}$), indicative of their proliferative phenotype (Supplementary Fig. 6). Other fetal-specific cell types included mesenchymal stem cells (*COL1A1^{lo}*, *TAGLN^{lo}*, *LUM^{hi}*), fibroblasts (*COL1A1^{hi}*, *TAGLN^{hi}*, *LUM^{lo}*)[39], endothelial cells (*PECAM1*)[40], and Hofbauer cells (*CD163*)[11] (Fig. 1b).

Fetal and maternal lymphocytes, B cells, and monocytes were also captured (Fig. 1b, c). We observed fetal and maternal B cells (*CD79A*)[41] and maternal plasma cells (*XBP1*, *IGHA* and other

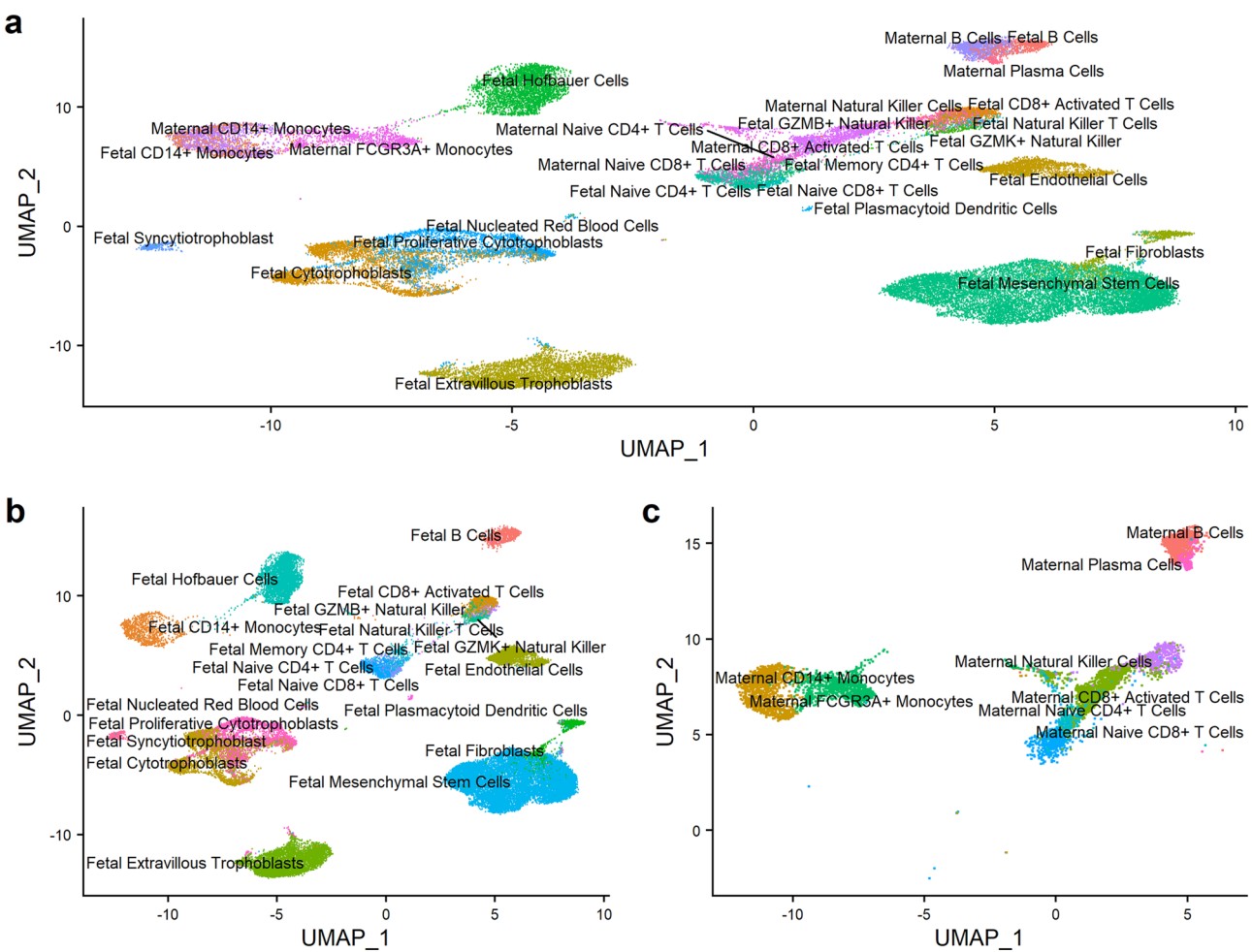

**Fig. 1 Integrated single-cell gene expression map of healthy placental villous tissue. a** Uniform Manifold Approximation and Projection (UMAP) plot of all cells ($n = 40,494$), with each cell colored by cell type cluster. **b** UMAP plot of fetal cells only ($n = 34,165$), with each cell colored by cell type cluster. **c** UMAP plot of maternal cells only ($n = 6329$), with each cell colored by cell type cluster.

immunoglobulins)[42]. We also observed fetal and maternal CD14+ monocytes ($CD14^+/FCGR3A^-$), maternal FCGR3A+ monocytes ($CD14^+/FCGR3A^+$)[43], and a small population of fetal plasmacytoid dendritic-like cells ($FLT3^+/ITM2C^+$)[44,45]. We further observed fetal and maternal natural killer cells ($NKG7$), fetal GZMB+ or GZMK+ natural killer cell subtypes, and fetal natural killer T cells ($NKG7^+/CD3E^+/CD8A^-$)[46,47]. Finally, we observed a variety of T cell subtypes: naïve CD4+ ($CCR7$, $CD3E$, $CD4$), naïve CD8+ ($CCR7$, $CD3E$, $CD8A$), memory CD4+ ($S100A4$, $CD3E$, $CD4$, $IL2$, $CCR7^{lo}$), and activated CD8+ T cells ($NKG7$, $CD3E$, $CD8A$) (Supplementary Fig. 5b)[48].

To identify upregulated genes in each cell type, we compared the expression of a gene in one cell type against that gene's average expression in all other cell types (Supplementary Data 2). Consequently, the same genes could be upregulated across several cell types of a similar lineage. $FLT1$ expression was highly upregulated in extravillous trophoblasts (log$_2$ fold-change (FC) = 3.89, $p_{adj} < 0.001$). Trophoblast cell types had the largest and most diverse transcriptomes, characterized by the largest number of unique RNA transcripts and detected genes per cell (Supplementary Fig. 7). Functional analysis of upregulated genes revealed cell type-specific biological processes (Supplementary Data 3). For example, fetal extravillous trophoblasts were enriched for genes relevant to placental structure and function such as cell migration ($p_{adj} < 0.001$) and response to oxygen levels ($p_{adj} < 0.001$) and syncytiotrophoblasts were enriched for genes

involved in steroid hormone biosynthetic process ($p_{adj} < 0.001$). Technical replication in Michigan samples 1 and 2 appeared high in UMAP space (Supplementary Fig. 8a, b). Indeed, the average intra-cluster gene expression between technical replicates had an average Spearman correlation (mean ± standard deviation) of $0.94 ± 0.14$ for sample 1 and $0.88 ± 0.20$ for sample 2 ($p$ values < 0.001).

**Single-cell RNA-sequencing deconvolution reference exhibits excellent in silico performance.** Based on the single-cell data, we created a placental signature gene matrix that incorporated expression information across an algorithmically selected 5229 signature genes to estimate the cellular composition of 27 fetal and maternal cell types from whole tissue gene expression data (Supplementary Fig. 9). To test the performance and robustness of this placental single-cell RNA-sequencing deconvolution reference, we randomly split our analytic single-cell RNA-sequencing dataset into 50% training and 50% testing subsets with balanced cell type proportions[49]. The same training dataset was used for each comparison; test mixtures were generated from the testing half of the dataset. Using a signature gene expression matrix generated from the training data, we estimated cell type composition in in silico pseudo-bulk testing data mixtures of known cell type composition with varying contributions of fetal vs. maternal origin cells and male vs. female fetal cells

**Table 1 Number of cells captured by single-cell RNA-sequencing in the final analytic dataset for each cell type by sample source.**

Distribution of cell types by sample and fetal/maternal origin

| Cell type | 1A | 1B | 2A | 2B | 3 | 4 | 5 | 6 | 7 | 8C | 8P | 9C | 9P | Count | Overall proportions | Fetal origin | Maternal origin |
|---|---|---|---|---|---|---|---|---|---|---|---|---|---|---|---|---|---|
| Fetal B cells | 109 | 149 | 195 | 229 | 38 | 27 | 15 | 0 | 35 | 9 | 1 | 1 | 1 | 809 | 2.0% | 2.4% | - |
| Fetal CD14+ monocytes | 232 | 187 | 174 | 197 | 45 | 23 | 42 | 1 | 11 | 3 | 0 | 1 | 0 | 916 | 2.3% | 2.7% | - |
| Fetal CD8+ activated T cells | 220 | 205 | 148 | 162 | 78 | 76 | 6 | 0 | 29 | 1 | 2 | 0 | 12 | 939 | 2.3% | 2.7% | - |
| Fetal cytotrophoblasts | 3 | 7 | 0 | 0 | 350 | 859 | 841 | 261 | 374 | 82 | 203 | 163 | 359 | 3502 | 8.6% | 10.3% | - |
| Fetal endothelial cells | 6 | 11 | 22 | 15 | 1 | 4 | 3 | 1058 | 180 | 181 | 84 | 236 | 137 | 1938 | 4.8% | 5.7% | - |
| Fetal extravillous trophoblasts | 0 | 0 | 0 | 0 | 6 | 222 | 3 | 234 | 3336 | 251 | 97 | 5 | 2 | 4156 | 10.3% | 12.2% | - |
| Fetal fibroblasts | 1 | 0 | 0 | 3 | 25 | 17 | 50 | 335 | 246 | 179 | 46 | 75 | 51 | 1028 | 2.5% | 3.0% | - |
| Fetal GZMB+ natural killer | 10 | 7 | 7 | 7 | 48 | 16 | 6 | 1 | 9 | 3 | 1 | 0 | 4 | 119 | 0.3% | 0.3% | - |
| Fetal GZMK+ natural killer T cells | 7 | 13 | 7 | 11 | 106 | 90 | 10 | 0 | 4 | 2 | 0 | 0 | 0 | 250 | 0.6% | 0.7% | - |
| Fetal Hofbauer cells | 23 | 27 | 115 | 136 | 107 | 114 | 266 | 240 | 607 | 932 | 326 | 49 | 174 | 3316 | 7.7% | 9.1% | - |
| Fetal memory CD4+ T cells | 26 | 41 | 42 | 45 | 68 | 54 | 28 | 11 | 8 | 3 | 1 | 5 | 0 | 332 | 0.8% | 1.0% | - |
| Fetal mesenchymal stem cells | 53 | 46 | 54 | 60 | 377 | 11 | 501 | 1207 | 4127 | 1806 | 606 | 1633 | 1755 | 12,236 | 30.2% | 35.8% | - |
| Fetal naive CD4+ T cells | 222 | 215 | 225 | 213 | 30 | 55 | 25 | 10 | 62 | 5 | 6 | 7 | 2 | 1077 | 2.7% | 3.2% | - |
| Fetal naive CD8+ T cells | 47 | 30 | 100 | 117 | 17 | 37 | 9 | 1 | 12 | 0 | 4 | 0 | 0 | 374 | 0.9% | 1.1% | - |
| Fetal natural killer T cells | 30 | 34 | 22 | 32 | 51 | 42 | 8 | 0 | 1 | 1 | 0 | 0 | 0 | 221 | 0.5% | 0.6% | - |
| Fetal nucleated red blood cells | 0 | 0 | 1 | 1 | 10 | 3 | 0 | 2 | 14 | 1 | 1 | 1 | 0 | 34 | 0.1% | 0.1% | - |
| Fetal plasmacytoid dendritic cells | 3 | 1 | 2 | 7 | 21 | 12 | 7 | 11 | 16 | 4 | 3 | 10 | 7 | 104 | 0.3% | 0.3% | - |
| Fetal proliferative cytotrophoblasts | 1 | 0 | 1 | 0 | 111 | 185 | 282 | 334 | 680 | 141 | 211 | 231 | 453 | 2630 | 6.5% | 7.7% | - |
| Fetal syncytiotrophoblast | 7 | 18 | 6 | 3 | 6 | 73 | 0 | 34 | 133 | 3 | 9 | 54 | 38 | 384 | 0.9% | 1.1% | - |
| Maternal B cells | 118 | 144 | 347 | 339 | 2 | 22 | 0 | 0 | 6 | 2 | 2 | 0 | 0 | 982 | 2.4% | - | 15.5% |
| Maternal CD14+ monocytes | 271 | 282 | 181 | 212 | 21 | 6 | 8 | 0 | 122 | 3 | 1 | 0 | 1 | 1108 | 2.7% | - | 17.5% |
| Maternal CD8+ activated T cells | 449 | 435 | 266 | 277 | 33 | 47 | 4 | 0 | 41 | 29 | 6 | 20 | 2 | 1609 | 4.0% | - | 25.4% |
| Maternal FCGR3A+ monocytes | 97 | 114 | 40 | 48 | 31 | 9 | 78 | 25 | 521 | 67 | 40 | 11 | 10 | 1091 | 2.7% | - | 17.2% |
| Maternal naive CD4+ T cells | 52 | 42 | 56 | 66 | 11 | 20 | 8 | 0 | 31 | 7 | 1 | 3 | 1 | 298 | 0.7% | - | 4.7% |
| Maternal naive CD8+ T cells | 74 | 95 | 181 | 172 | 8 | 41 | 5 | 0 | 11 | 0 | 0 | 0 | 0 | 587 | 1.4% | - | 9.3% |
| Maternal natural killer cells | 114 | 139 | 68 | 60 | 19 | 8 | 3 | 0 | 56 | 10 | 0 | 3 | 1 | 481 | 1.2% | - | 7.6% |
| Maternal plasma cells | 39 | 38 | 32 | 46 | 0 | 8 | 2 | 0 | 7 | 1 | 0 | 0 | 0 | 173 | 0.4% | - | 2.7% |
| Count | 2214 | 2280 | 2292 | 2458 | 1620 | 2081 | 2210 | 3765 | 10,679 | 3726 | 1651 | 2508 | 3010 | 40,494 | 100.0% | 34,165 | 6329 |

Overall cell composition by cell count provided for each cell type. Proportions represent the overall proportion of that cell type in the dataset or among cells of only fetal or maternal origin. The final analytic sample included 40,494 cells and 36,601 genes across nine biological replicates, two of which had a technical replicate (Samples 1B and 2B) and another two included peripheral subsampling (Samples 8P and 9P).

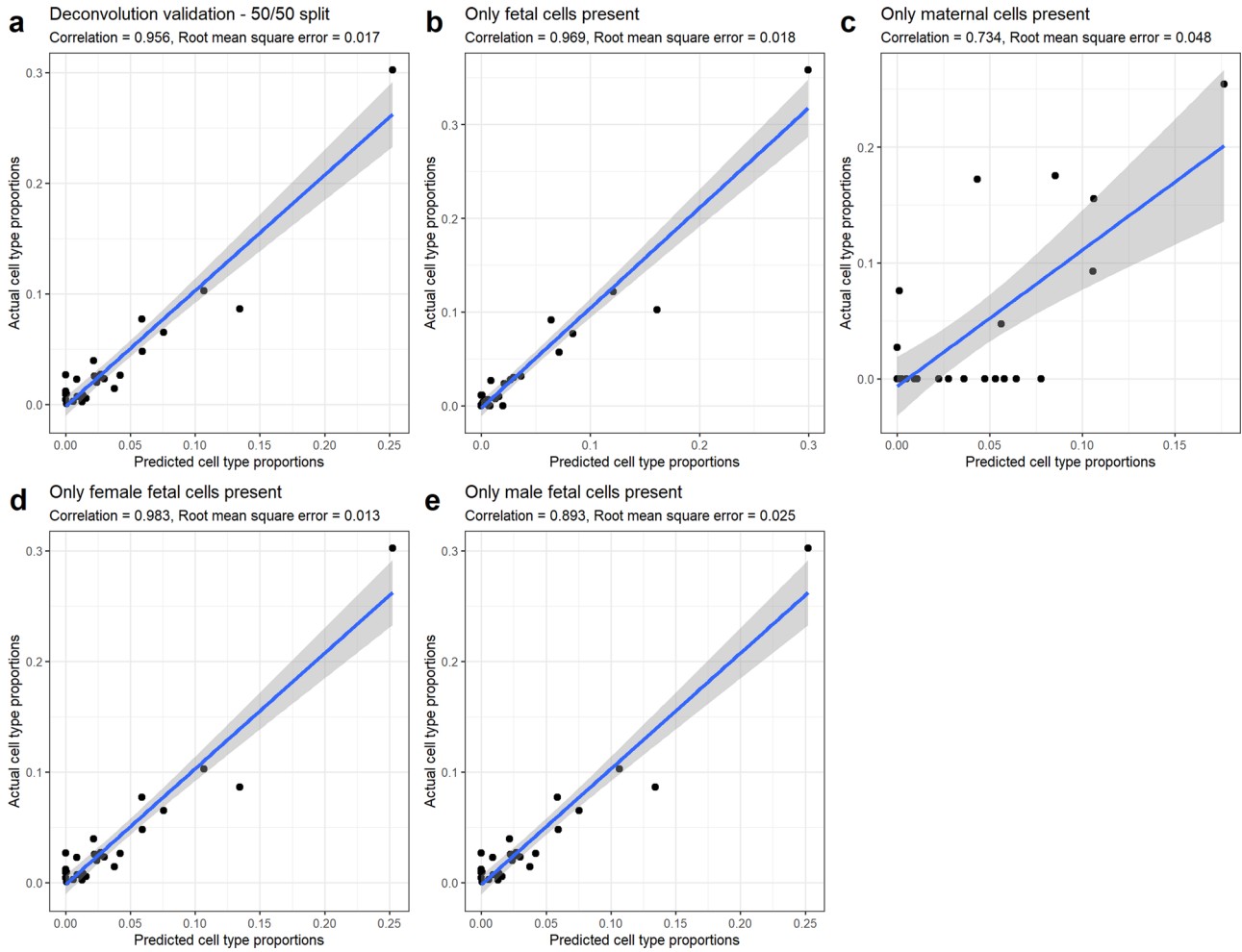

**Fig. 2 In silico placental deconvolution testing.** Scatter plots summarizing the performance of our single-cell deconvolution reference using CIBERSORTx with in silico mixtures of single-cell libraries from a 50/50 training/test split of the integrated single-cell RNA-seq dataset ($n = 40,494$). The same training dataset was used for each comparison; test mixtures were generated from the testing half of the dataset. Predicted deconvoluted cell type proportions for each of the 27 cell types are encoded on the x-axis. Actual cell type proportions from the test dataset are encoded on the y-axis. Correlation coefficients and root mean square error measures are presented for each comparison. A linear line of best fit overlays the results. The gray shaded area represents the 95% confidence intervals around the simple linear regression estimates. **a** The test mixture is the test half of the single-cell dataset ($n = 20,242$). **b** The test mixture sampled only fetal cells ($n = 17,080$). **c** The test mixture sampled only maternal cells ($n = 3162$). **d** The test mixture sampled only female fetal cells ($n = 8394$). **e** The test mixture sampled only male fetal cells ($n = 8394$).

(Fig. 2). In all mixtures, the 27 predicted and actual cell type proportions were correlated ($p$ value $< 0.001$ for each test). In the primary deconvolution analysis of all cell types at their natural rates ($n = 20,242$), estimated and actual cell type proportions had a Pearson correlation coefficient of 0.956 (95% CI [0.904, 0.980]). The worst performance was under the unrealistic scenario that the mixture was composed entirely of maternal cell types ($n = 3162$) with a Pearson correlation of 0.734 (95% CI [0.491, 0.871]) between actual estimated cell type proportions. Our deconvolution reference was also robust to fetal sex when only male fetal cells (Pearson correlation $= 0.893$, 95% CI [0.776, 0.950]) were included ($n = 8394$), or only female fetal cells (Pearson correlation $= 0.983$, 95% CI [0.964, 0.993]) ($n = 8394$). Together, these results show that our reference panel can successfully deconvolute placental tissues, though some maternal cell types common to both mother and fetus may be erroneously labeled fetal in the absence of fetal cells of those cell types.

**Fluorescence-activated cell sorting of major placental cell types yielded mixed cell type isolation results.** We isolated whole bulk placental villous tissue, enriched syncytiotrophoblasts, and sorted

five cell types (Hofbauer cells, endothelial cells, fibroblasts, leukocytes, extravillous trophoblasts, and cytotrophoblasts) via FACS from four healthy term, uncomplicated Cesarean sections for bulk RNA-sequencing, labeled Sorted 1 (same sample source as single-cell RNA-sequencing sample 1), Sorted 2, Sorted 3, and Sorted 4 (Supplementary Fig. 10 and Supplementary Table 2). For analysis, as recommended[50], we excluded 19,048 genes that were not present in at least 3 samples and an additional 865 genes that did not have a cumulative library size-normalized count of at least 10. Principal component (PC) analysis of whole-transcriptome sorted-cell bulk RNA-sequencing normalized counts is provided in Supplementary Fig. 11.

To identify upregulated genes in each cell type, we compared the expression of a gene in one cell type against that gene's average expression in all other cell types (Supplementary Fig. 12). Consequently, the same genes could be upregulated across several cell types of a similar lineage. All 38,468 uniquely mapping genes were tested. A total of 746 genes were algorithmically dropped from the syncytiotrophoblast contrast due to excessively low counts, low variability, or extreme outlier status. Large-scale gene expression differences were observed for each cell type (Supplementary Data 4).

Functional analysis of upregulated genes revealed cell type-specific biological processes (Supplementary Data 5). For example, syncytio-trophoblasts were enriched for genes relevant to placental structure and function such as angiogenesis, cell-substrate adhesion, and regulation of epithelial cell proliferation ($p_{adj} < 0.001$). To compare sorted and single-cell differential expression and enrichment results, we tabulated the number of unique genes and pathways overlapping between the two analyses after collapsing the single-cell cell type cluster labels to the seven cell type fractions that we had targeted to isolate for downstream analyses (Supplementary Table 3). On average, 15.0% of single-cell upregulated genes and 5.9% of enriched pathways were also identified among the sorted-cell data. On average, 17.5% of sorted cell type upregulated genes and 39.2% of pathways were also identified among the single-cell data. Sorted endothelial cell results were limited due to the limited number of biological replicates.

We applied the single-cell deconvolution reference to estimate cell proportions in the 4 whole tissue (with 1 additional technical replicate) and 19 sorted or enriched cell type fractions. We collapsed the single-cell cell type cluster labels to the seven cell type fractions we targeted for isolation for downstream analyses (Supplementary Data 6, Sheet 1). All deconvoluted samples exhibited high goodness-of-fit between original bulk mixtures and the estimated cell type proportion mixtures ($p$ values < 0.001). Among the signature genes, original bulk and estimated cell type fractions had a Pearson correlation (mean ± standard deviation) of 0.73 ± 0.11 and root mean square error of 0.88 ± 0.04 (Supplementary Data 6, Sheet 2). Deconvolution results (mean ± standard deviation) suggest we successfully isolated fibroblast- ($n = 3$, 74.7% ± 0.6%) and leukocyte-enriched ($n = 4$, 82.3% ± 24.8%) cell type fractions. Other cell type targets were less successful (range 0–26% estimated purity). The Hofbauer cell fraction was predicted to be mostly leukocytes ($n = 4$, 91.5% ± 0.5%).

**Cell proportion deconvolution of bulk placental tissue pre-eclampsia dataset.** We applied the single-cell deconvolution reference to estimate cell proportions from bulk placental tissue in 157 preeclampsia cases and 173 controls[33] compiled from eight previously published studies[33,51–57]. Mean gestational age was 2.2 weeks younger in cases than controls ($p$ value < 0.001, Table 2). All deconvoluted samples exhibited high goodness-of-fit between original bulk mixtures and the estimated cell type pro-portion mixtures ($p$ values < 0.001). Among the signature genes, original bulk and estimated mixtures had a Pearson correlation (mean ± standard deviation) of 0.70 ± 0.04 and root mean square error of 0.73 ± 0.03 (Supplementary Data 7). Fetal naïve CD4+ T cells and fetal GZMB+ natural killer cells were estimated to be at 0% abundance in all samples and were dropped from down-stream analyses. Cytotrophoblasts were the most abundant (mean ± standard deviation) estimated fetal cell type (27.9% ± 4.3%) followed by syncytiotrophoblasts (23.4% ± 5.0%) and mesenchymal stem cells (10.3% ± 3.3%). The most common maternal cell types were naïve CD8+ T cells (2.8% ± 1.5%), plasma cells (1.4% ± 0.7%), and B cells (1.3% ± 0.8%). A com-parison of deconvoluted whole tissue cell type proportions among healthy individuals (Supplementary Fig. 13) between the micro-array dataset GSE75010 ($n = 173$ controls), our whole tissue bulk RNA-sequencing samples (sorted samples 1–4), and the single-cell dataset compiled here (single-cell samples 1–9) suggests that syncytiotrophoblasts and endothelial cells are underrepresented in the single-cell data. This is likely due to dissociation bias, which has been commonly observed in single-cell assays of other tissues[58]. Overall, the Pearson correlation of the average decon-voluted cell type proportion across the 27 cell types between healthy bulk RNA-sequencing and microarray controls was 0.80 (95% CI: [0.60, 0.91]).

**Table 2 Demographic characteristics of eight previously published bulk microarray placental gene expression case–control studies (accessed through GSE75010) for deconvolution application testing.**

| Descriptive statistics of microarray preeclampsia case–control studies | | | |
|---|---|---|---|
| | **Control (N = 173)** | **Preeclampsia (N = 157)** | **P value** |
| Fetal sex | | | |
| Female | 78 (45.1%) | 81 (51.6%) | 0.28 |
| Male | 95 (54.9%) | 76 (48.4%) | |
| Gestational age (weeks) | | | |
| Mean (SD) | 35.2 (3.97) | 33.0 (3.17) | <0.001 |
| Median [min, max] | 37.0 [25.0, 41.0] | 33.0 [25.0, 39.0] | |
| Study | | | |
| GSE10588 | 26 (15.0%) | 17 (10.8%) | 0.39 |
| GSE24129 | 8 (4.6%) | 8 (5.1%) | |
| GSE25906 | 37 (21.4%) | 23 (14.6%) | |
| GSE30186 | 6 (3.5%) | 6 (3.8%) | |
| GSE43942 | 7 (4.0%) | 5 (3.2%) | |
| GSE44711 | 8 (4.6%) | 8 (5.1%) | |
| GSE4707 | 4 (2.3%) | 10 (6.4%) | |
| GSE75010 | 77 (44.5%) | 80 (51.0%) | |

Bivariate batch with Kruskal–Wallis ANOVA (regular ANOVA homogeneity of variances violated) for continuous variables and $\chi^2$ test for categorical outcomes.

**Differentially abundant cell type proportions in preeclampsia cases versus controls.** To test for differences in cell proportions between preeclampsia cases and controls (Supplementary Fig. 14), we fit beta regression models for each cell type proportion adjusted for study source, fetal sex, and gestational age to estimate the prevalence odds ratio for each cell type (Supplementary Data 8). Among fetal cell types, extravillous trophoblasts ($p < 0.001$), memory CD4+ T cells ($p = 0.007$), CD8+ activated T cells ($p = 0.005$), and natural killer T cells ($p = 0.006$) were more abundant (Fig. 3) in preeclampsia cases relative to controls. The unadjusted median extravillous trophoblast abundance was 6.4% among cases compared to 2.1% among controls. Mesenchymal stem cells (median percent composition in cases vs. controls, 8.8% vs. 11.0%), Hofbauer cells (2.7% vs. 4.4%), and fetal naïve CD8+ T Cells (4.2% vs. 4.5%) were all less abundant among preeclampsia cases compared to controls ($p < 0.001$). Among maternal cell types, maternal plasma cells (1.6% vs. 1.2%) were more abundant among preeclampsia cases compared to controls ($p < 0.001$).

**Differential expression between preeclampsia cases and controls attenuated by cell type proportion adjustment.** To test whether microarray gene expression differences between pre-eclampsia cases and controls are partly driven by differences in cell type abundances, we fit linear differential gene expression models adjusted for covariates study source, fetal sex, and gestational age with and without adjustment for deconvoluted cell type propor-tions. To reduce the number of model covariates and account for dependence between deconvoluted cell type proportions, we applied PC analysis to deconvoluted cell type proportions. The first five PCs accounted for 87.2% of the variance in decon-voluted cell type proportions and were added as additional cov-ariates to form the cell type-adjusted model. Variation in PCs 1 and 2 was largely driven by syncytiotrophoblasts (33.8%), extra-villous trophoblasts (33.5%), and cytotrophoblasts (15.3%) pro-portions and provided some separation between cases from controls (Supplementary Fig. 15a, c). Variation in PC3 was largely driven by cytotrophoblasts (50.1%) and to a lesser extent

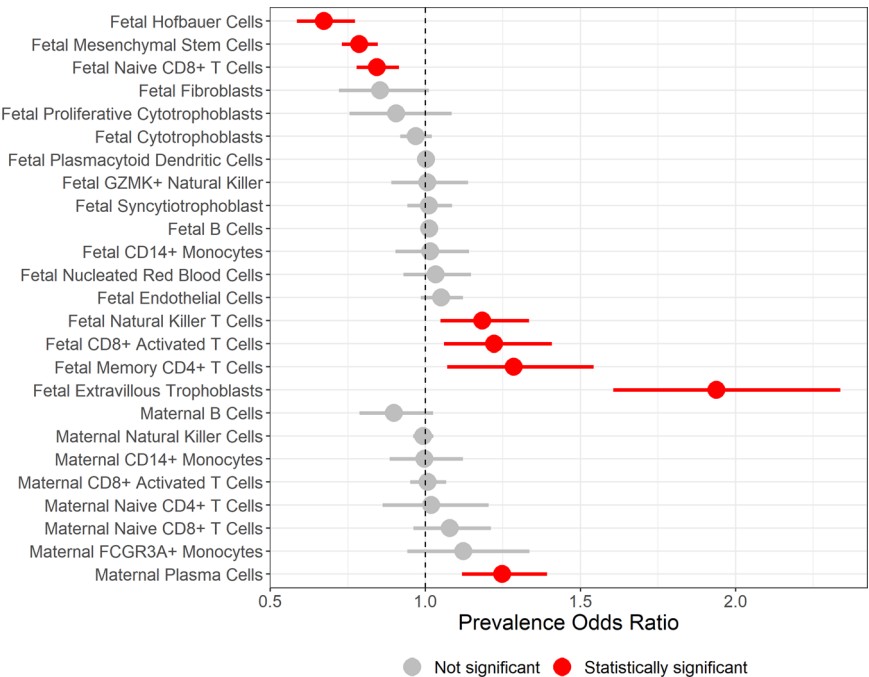

**Fig. 3 Preeclampsia case–control status and cell type proportion differential abundance analysis.** Forest plot of multivariate beta regression models' prevalence odds ratio estimates adjusted for study source, gestational age, and fetal sex tested for a difference in each cell type's proportions in cases versus controls (n = 157 cases, 173 controls). Horizontal lines indicate the range of the 95% confidence interval.

syncytiotrophoblasts (16.6%), mesenchymal stems cells (14.5%), and extravillous trophoblasts (13.7%) (Supplementary Fig. 15b, d).

In the cell type-naïve base models (n = 14,651 genes, 173 controls, and 157 cases) adjusted for study source, gestational age, and fetal sex, 550 genes were differentially upregulated and 604 were downregulated in preeclampsia cases versus controls (Fig. 4a and Supplementary Data 9). Gene set enrichment analysis of biological processes identified 41 overrepresented pathways in the base model (Fig. 5a and Supplementary Data 10). Biological process pathways such as aerobic respiration (false discovery adjusted $q < 0.001$), mitochondrial respiratory chain complex assembly ($q < 0.001$), glutathione metabolism ($q = 0.003$), and ribosome biogenesis ($q = 0.001$) were overrepresented among downregulated genes. No pathways were overrepresented among upregulated genes, though intermediate filament organization ($q = 0.26$), keratinocyte differentiation ($q = 0.77$), and endothelial cell development ($q = 0.42$) had comparable enrichment scores. Remarkably, when the base model was additionally adjusted for the first five PCs of imputed cell type proportions, there were zero differentially expressed genes between preeclampsia cases and controls (Fig. 4b and Supplementary Data 9). Of the cell type-adjusted results, 19 pathways were overrepresented (Fig. 5b and Supplementary Data 10). Downregulation of mitochondrial respiratory chain complex assembly ($q < 0.001$), aerobic respiration ($q = 0.001$), ribosome biogenesis ($q = 0.001$), and glutathione metabolism ($q = 0.02$) were overrepresented among downregulated genes. Detection of chemical stimulus involved in sensory perception of smell ($q = 0.04$) and non-coding RNA processing ($q = 0.04$) were also overrepresented pathways among downregulated genes. Neuroepithelial cell differentiation ($q = 0.04$) was overrepresented among upregulated genes. Vascular endothelial growth factor receptor signaling pathway ($q = 0.15$), of which *FLT1* is a member, had an enrichment score of 1.77 (up from 1.34, $q = 0.43$ in the base model) but did not meet the $q$ value cutoff. Overall, downregulation of mitochondrial biogenesis, aerobic respiration, and ribosome biogenesis and related pathways were robust to cell type proportion adjustment.

**Differential expression of preeclampsia-associated genes mediated by placental cell type proportions.** Overexpression of *FLT1* in placental tissue[59–62], detection of a soluble isoform of *FLT1* in maternal circulation[63,64], and fetal genetic variants near *FLT1*[65] have implicated *FLT1* in preeclampsia etiology. Because we observed cell type-specific expression patterns of *FLT1* in trophoblasts, particularly in extravillous trophoblasts, we hypothesized that the observed attenuation of *FLT1* differential expression may be due in part to the differences in cell type proportions observed between preeclampsia cases and controls. To test this hypothesis, we applied a unified mediation and interaction analysis to quantify the proportion of *FLT1* expression differences mediated by deconvoluted cell type proportions. We did not observe an interaction between preeclampsia status and cellular composition (overall proportion attributable to interaction = −5.8%, 95% CI [−17.1%, 5.0%]). We therefore dropped interaction parameters from the model for the final analysis. In the model without interaction, 37.8% (95% CI [27.5%, 48.8%]) of the 1.05 (95% CI [0.89, 1.21]) log$_2$ signal intensity increase in the association between preeclampsia and *FLT1* expression was attributable to differences in placental cell composition between preeclampsia cases and controls (Fig. 6). Overexpression of *LEP* and *ENG* have also been associated with preeclampsia[59–62]. Mediation results were similar for *LEP* (total effect = 2.62 (95% CI [2.26, 2.97] log$_2$ signal intensity increase; proportion mediated = 34.5% (95% CI [26.0%, 44.9%]) and *ENG* (total effect = 0.93 (95% CI [0.79, 1.07] log$_2$ signal intensity increase; proportion mediated = 34.5% (95% CI [25.0%, 45.3%]).

## Discussion

To create the largest publicly available placental RNA deconvolution reference of 19 fetal and 8 maternal cell type-specific gene expression profiles, we newly sequenced placental villous cells, integrated those results with data from previously published studies, and built a signature gene matrix for deconvolution of bulk villous tissue gene expression data. In silico testing of our deconvolution reference demonstrated successful and robust

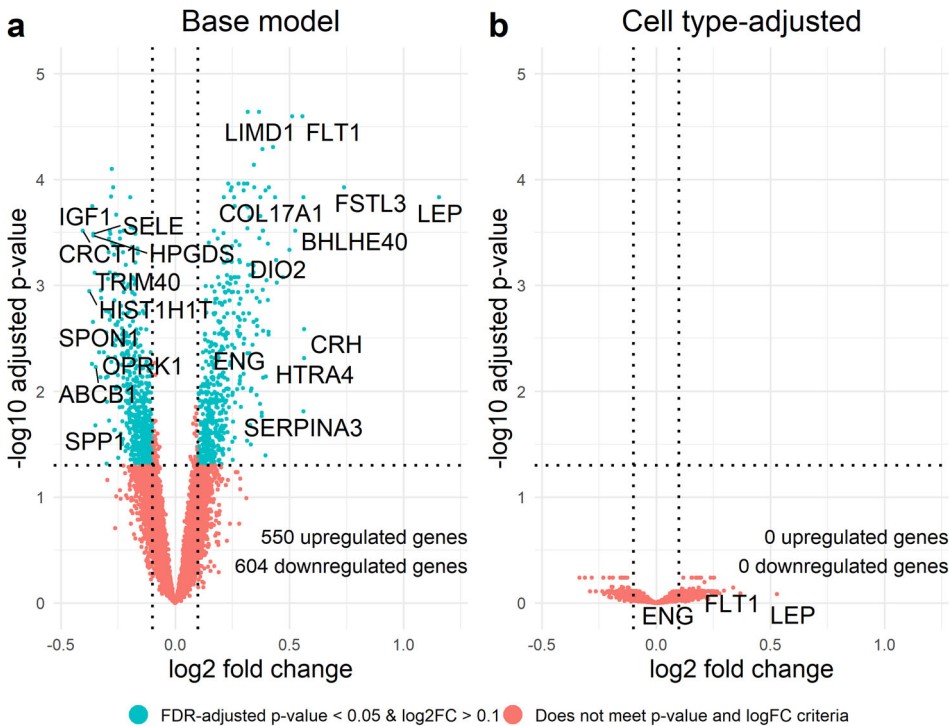

**Fig. 4 Preeclampsia case–control differential expression analysis.** Volcano plots comparing differentially expressed genes in samples from 153 preeclampsia cases versus 173 healthy controls across two models: **a** the base model adjusted for covariates fetal sex, study source, and gestational age and **b** the model adjusted for fetal sex, study source, and gestational age and additionally adjusted for the first five principal components of estimated cell type proportions. Dotted line represents a false discovery rate-adjusted *q* value of 0.05. *FLT1*, *LEP*, and *ENG* are labeled as genes of interest in preeclampsia.

deconvolution. To compare single-cell placental cell type expression profiles to more conventional sorting methods, we created a FACS scheme to enrich and sequence RNA from five important placental cell types as well as syncytiotrophoblasts. Deconvolution of sorted cell type fractions with the single-cell deconvolution reference suggested most conventionally sorted cell types are far less pure than what can be accomplished with clustering and aggregation of single-cell results, and at much lower cell type resolution. We applied the single-cell deconvolution reference to estimate cell type proportions in a previously published epidemiologic microarray study of the pregnancy complication preeclampsia, revealing placental cell type proportion differences between preeclampsia cases and controls at term. We then showed that large gene expression differences between preeclampsia cases and controls were markedly attenuated after adjustment for cell type proportions. Preeclampsia-associated pathways, including downregulation of mitochondrial biogenesis, aerobic respiration, and ribosome biogenesis were robust to cell type adjustment, suggesting direct changes to these pathways. Finally, to quantify the attenuation of differential expression of the preeclampsia biomarkers *FLT1*, *LEP*, and *ENG*, we applied mediation analysis to show cellular composition mediated a substantial proportion of the association between preeclampsia and *FLT1*, *LEP*, and *ENG* overexpression. Cell type proportions may be an important and often overlooked factor in gene expression differences in placental tissue studies.

By integrating our new single-cell RNA-sequencing results with those from a previously published study, our integrated dataset, to our knowledge, is the largest and possibly only reference available for healthy, term placental villous tissue to date. We document term cell type-specific gene expression patterns for well-characterized placental cell types, including syncytiotrophoblasts[10], cytotrophoblasts[13], and extravillous trophoblasts[14]. In addition, we provide gene expression

markers for relatively understudied placental cell types such as endothelial cells, mesenchymal stem cells, and Hofbauer cells as well as maternal peripheral mononuclear cells recovered from the maternal-fetal interface. Compared to the previous analysis of the published samples[17] which relied on predominately sex-specific gene expression markers to differentiate proliferative from non-proliferative cytotrophoblasts, we show that functional enrichment analysis revealed broad upregulation of proliferation pathways in proliferative cytotrophoblasts. The low representation of some cell types such as trophoblasts in our single-cell RNA-sequencing results from the Michigan study suggests that these cell types may be especially sensitive to dissociation and disintegrate before transcript capture, commonly referred to as dissociation bias[58]. Michigan samples 1 and 2 also included a cryopreservation step like those employed in large-scale epidemiological studies that may have exacerbated dissociation bias[66]; this applies to both single-cell and sorted cell type experiments. Future studies may propose alternative approaches to perform unbiased single-cell RNA-sequencing in placental tissues; indeed, single-nucleus RNA-sequencing has been used to characterize an in vitro syncytiotrophoblast model and may be more appropriate to assay such cell types sensitive to dissociation procedures[67]. We verified that our deconvolution reference exhibited strong performance even with extremely imbalanced and unlikely real-world test mixture distributions by fetal sex and maternal cell type representation.

Our preeclampsia findings are consistent with a prior patho-physiological understanding of the disorder, linking cell type proportion estimates and gene expression data in bulk tissue. Among preeclampsia cases, we observed an elevated proportion of extravillous trophoblasts and underrepresentation of stromal cell types, which may reflect an arrest in conventional placental cell type differentiation and maturation following insufficient uterine spiral artery remodeling implicated in preeclampsia[68–70].

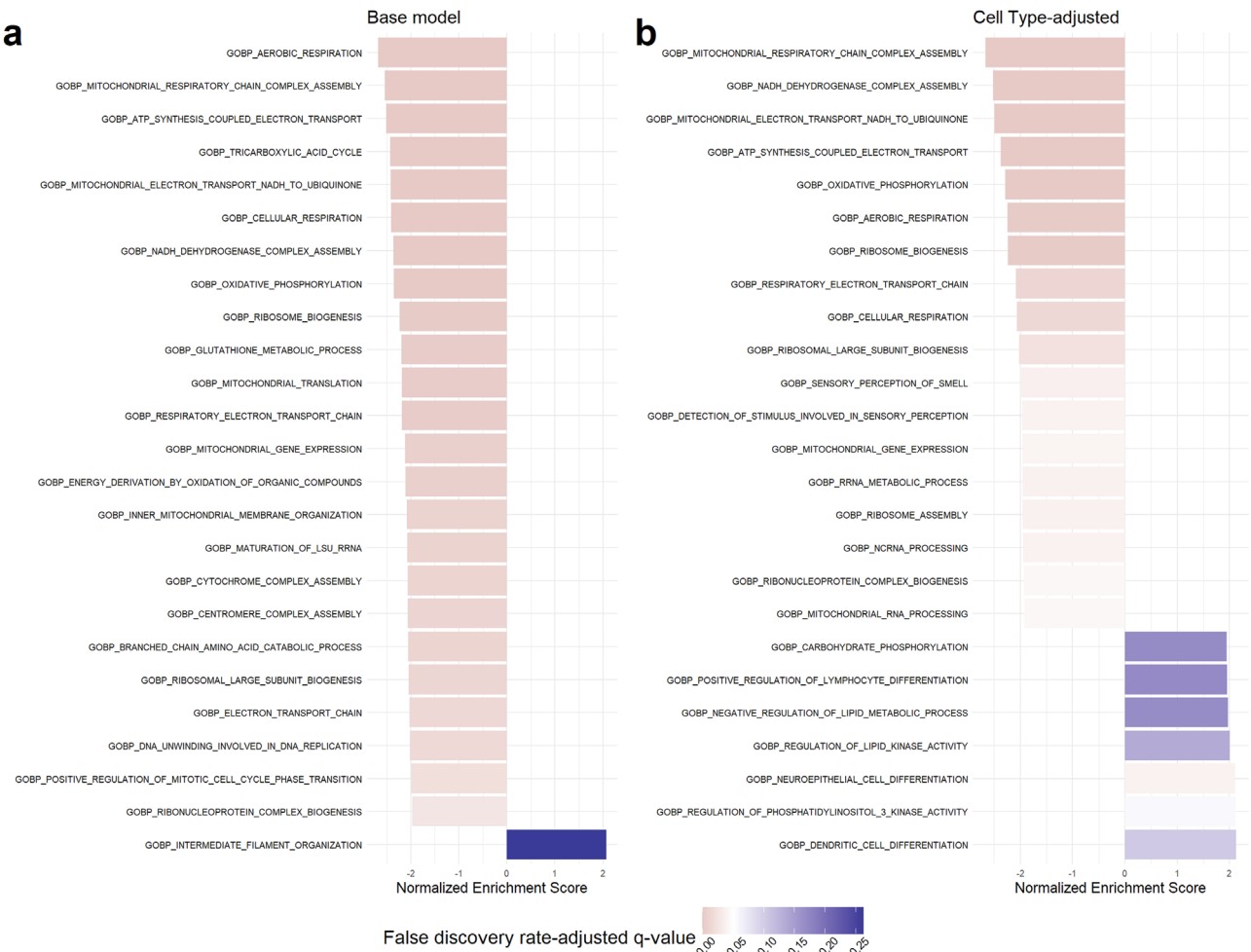

**Fig. 5 Preeclampsia case–control differential expression enrichment analysis.** Top Gene Set Enrichment Analysis pathways from the Gene Ontology: Biological Processes database results for the differential expression analysis by preeclampsia case–control status. Results arranged by descending magnitude of the absolute value of the normalized enrichment score. Pathways colored red are significant at a false discovery rate-adjusted (FDR) $q$ value of 0.05 whereas pathways in blue are statistically insignificant. **a** Top pathways from the cell type-unadjusted analysis. **b** Top pathways from the cell type-adjusted analysis.

A recent study of bulk placental gene expression across trimesters suggests that Hofbauer cells may more abundant in the second trimester compared to the third, possibly to support vasculo-genesis, though this study involved a small deconvolution reference that contained a limited variety of cell types[71]. A better understanding of the evolution of temporal placental composition changes may yield greater insight into placental pathologies. In the cell type-naïve differential expression model, consistent with previous findings, placentas from pregnancies with preeclampsia overexpressed *FLT1*, *LEP*, and *ENG*[59–62]. In our cell type-adjusted model, *FLT1* and *LEP* remained only nominally significant whereas *ENG* did not meet the nominal significance threshold. Mediation analysis confirmed that a significant proportion of *FLT1*, *LEP*, and *ENG* overexpression was attributable to differences in the cellular composition of the placenta. These results suggest that placental cell type proportion differences may be an overlooked factor in explaining the well-documented association between preeclampsia and *FLT1*, *LEP*, and *ENG* expression[59–62]. Downregulation of mitochondrial biogenesis, aerobic respiration, and ribosome biogenesis was robust to cell type adjustment, indicating direct changes to these pathways beyond shifts in cell type abundance. Indeed, disruption of the mitochondrial fission-fusion cycle[72], malperfusion[73,74], and inhibited protein synthesis secondary to endoplasmic reticulum stress[75,76] have all previously

been associated with preeclampsia. Interestingly, cell type adjustment increased the enrichment score results of vascular endothelial growth factor receptor signaling pathway, a mechanistic hypothesis in preeclampsia etiology[63,73,77,78], from 1.36 to 1.77 ($q = 0.43$ to $q = 0.15$). This approach may reveal the biological mechanisms of other diseases beyond cellular composition differences. Because oxygen tension is a critical factor in trophoblast differentiation, inappropriate oxygenation may partially explain the elevated proportion of extravillous trophoblasts, though regulators of this process such as *HIF1A* and *TGFB3*[79] were not differentially expressed at the tissue level in our analysis. A recent single-cell RNA-sequencing case–control study of pre-eclampsia, however, identified upregulation of *TGFB1* in extra-villous trophoblasts, potentially indicative of altered trophoblast differentiation or invasion[80,81]. A similar study revealed decreased activity of gene network modules regulated by transcription factors *ATF3*, *CEBPB*, and *GTF2B* and decreased expression of *CEBPB* and *GTF2B* in preeclamptic extravillous trophoblasts compared to controls; follow-up in vitro experiments suggested *CEBPB* and *GTF2B* knockdown reduced extra-villous trophoblast viability and invasion[82]. Consistent with our other findings, this study also observed a similar trend in cell type proportion differences and upregulation of *FLT1* in extravillous trophoblasts and *ENG* in syncytiotrophoblasts between

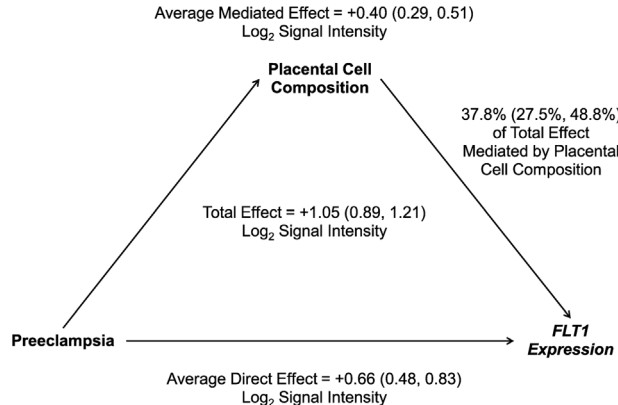

**Fig. 6 Placental cell composition as a mediator of *FLT1* expression.**
Mediation of *FLT1* gene expression by placental cell type composition
($n = 157$ cases, 173 controls). Placental cell composition was
operationalized as first five principal components of estimated cell type
proportions. 95% confidence intervals are provided after effect estimates
for each model parameter. The same framework was also applied with *LEP*
or *ENG* expression as the outcome.

preeclampsia cases and controls[80]. Future work should consider
and account for cell type proportions and the cell type-specific
expression patterns of genes that regulate placental development
or are associated with preeclampsia to better understand pre-
eclampsia etiology.

This study has several strengths. We profile the parenchymal
healthy term villous tissue in the placenta and integrate our dataset
with samples from previously published studies to generate the lar-
gest, to the best of our knowledge, cell type-specific placental villous
tissue gene expression reference to date. Single-cell RNA-sequencing
allowed us to agnostically capture diverse placental cell types without
a priori knowledge of cell types and their characteristics and tabulate
gene expression patterns at high resolution and specificity. Our in
silico deconvolution tests demonstrated robust performance to even
extreme distributions of maternal or sex of fetal cells. We demon-
strate technical replication of single-cell RNA-sequencing in placental
villous tissue. We were able to apply our findings to a large target
deconvolution dataset of preeclampsia that contained placental
measures from hundreds of participants across eight different studies.
Most importantly, we evaluate cell type proportion differences in an
epidemiological study of placental parenchymal tissue and pre-
eclampsia, and genome-wide gene expression differences accounting
for cell type heterogeneity, a critical limitation in bulk tissue assays.

This study also has several limitations. Although our cellular
sample size comprised of 40,494 cells is relatively large compared to
previous single-cell RNA-sequencing studies of term placental villous
tissue, this dataset still represents a limited biologic replicate sample
size compared to epidemiologic scale studies. Our newly sequenced
samples came from a convenience sample without available demo-
graphic information beyond uncomplicated and healthy Cesarean-
section status. Similarly, the sample size of FACS-sorted tissues was
limited, and some cell type fractions were excluded due to low RNA
quality or exhibited poor estimated purity, likely complicated by
degradation of cell surface markers from apoptosis characteristic of
development and parturition[83,84] and sample processing. This study
did not include placental tissues for single-cell analysis from pre-
eclamptic patients to confirm intra-cell type gene expression changes.
Despite excellent in silico performance, we had no external gold
standard to verify deconvolution performance. This deconvolution
reference may not be sensitive enough to discriminate between cell
subtypes such as proliferative vs. non-proliferative cytotrophoblasts
that are clearly delineated in the single-cell analysis; in such cases,
investigators may collapse cell type proportions counts into a single

major cell type group, such as cytotrophoblasts. Future studies may
verify whether cell type proportions estimated in diseased or vaginally
delivered tissues are robust to a deconvolution reference generated
from healthy villous tissue delivered via Cesarean-section. Residual
confounding may remain in our statistical models due to the limited
number of common covariates across all eight preeclampsia
case–control studies. Due to the nature of villous tissue sampling, our
study design is cross-sectional, limiting our ability to establish tem-
porality between exposure and outcome to rule out reverse causation.
As with any study conditioned on live birth, selection bias may affect
our results. However, the effects of harmful exposures that lead to
selection tend to be underestimated in these scenarios[85,86]. Therefore,
our results likely represent a conservative underestimate of the effects
of preeclampsia on inappropriate cell composition and preeclampsia
status on gene expression.

In summary, we provide a cell type-specific deconvolution
reference via single-cell RNA-sequencing in the parenchymal
placental term villous tissue. We demonstrated this reference was
robust to different distributions of maternal and fetal sex through
in silico validation testing. In addition, we benchmarked these
single-cell cell type-specific gene expression profiles against pla-
cental cell types isolated with more conventional FACS followed
by bulk RNA-sequencing. We applied this deconvolution refer-
ence to an epidemiologic preeclampsia dataset to reveal biologi-
cally relevant shifts in placental cell type proportions between
preeclampsia cases and controls. Once cell type proportion dif-
ferences were accounted for, differential gene expression differ-
ences were markedly attenuated between preeclampsia cases and
controls. Enrichment analysis revealed downregulation of mito-
chondrial biogenesis, aerobic respiration, and ribosome biogen-
esis were robust to cell type adjustment, suggesting direct changes
to these pathways. A substantial proportion of the overexpression
of the *FLT1*, *LEP*, and *ENG* in preeclampsia was mediated by
placental cell composition. These results add to the growing body
of literature that emphasizes the centrality of cell type hetero-
geneity in molecular measures of bulk tissues. We provide a
publicly available placental cell type-specific gene expression
reference for term placental villous tissue to overcome this critical
limitation.

## Methods

**Placental tissue collection and dissociation**. Placentas were collected shortly
after delivery from healthy, full-term, singleton uncomplicated Cesarean sections at
the University of Michigan Von Voigtlander Women's Hospital. Pregnant women
provided written informed consent for research use of discarded tissues. Study
protocols for discarded tissue collection and research use were approved by the
University of Michigan Institutional Review Board (HUM00017941,
HUM00102038). Villous placental tissue biopsies were collected and minced for
dissociation after cutting away the basal and chorionic plates and scraping villous
tissue from blood vessels[13]. We subjected approximately 1 g minced dissected
villous tissue to the Miltenyi Tumor Dissociation Kit on the GentleMACS Octo
Dissociator with Heaters (Miltenyi Biotec) to yield single-cell suspensions of viable
placental cells in 5 μM StemMACS™ Y27632 (Miltenyi Biotec) in RPMI 1640
(Gibco) according to manufacturer's instructions for soft tumor type. Red blood
cells were depleted using RBC lysis buffer (BioLegend) according to manufacturer's
protocol A. Single-cell suspensions were size-filtered at 100 μm to remove undi-
gested tissue and subsequently at 40 μm[12,13]. To collect a syncytiotrophoblast-
enriched fraction, the fraction between 40 and 100 μm was washed from the 40-μm
strainers, adapting a previous protocol that collected syncytiotrophoblasts
throughout this size range[10]. Single-cell suspensions <40 μm were cryogenically
stored in 5 μM StemMACS™ Y27632 90% heat-inactivated fetal bovine serum
(Gibco)/10% dimethyl sulfoxide (Invitrogen). For each placenta, additional whole
villous tissue samples were stored in RNALater (Qiagen).

Previously published single-cell RNA-sequencing raw data of healthy, term
placental villous tissue samples came from the Database of Genotypes and
Phenotypes (Pique-Regi et al., accession number phs001886.v1.p1[87]) SRR10166478
(Sample 3), SRR10166481 (Sample 4), and SRR10166484 (Sample 5)[17]. The
collection and use of human materials for the study were approved by the
Institutional Review Boards of the Wayne State University School of Medicine. All
participating women provided written informed consent prior to sample
collection[17]. Additional previously published samples came the European Genome-

Phenome Archive (Tsang et al., accession number EGAS00001002449[88]) (Samples 6–9)[16]. The study was approved by the Joint Chinese University of Hong Kong-New Territories East Cluster Clinical Research Ethics Committee, and informed consent was obtained after the nature and possible consequences of the studies were explained. Pregnant women were recruited from the Department of Obstetrics and Gynecology, Prince of Wales Hospital, Hong Kong with informed consent; the subjects studied had consented to sequencing data archiving[16].

**Placental single-cell RNA-sequencing**. Villous tissue single-cell suspensions were thawed and sorted via FACS with LIVE/DEAD Near-IR stain (Invitrogen) for viability and forward-scatter and side-scatter profiles to eliminate cellular debris and cell doublets. Viability- and size-sorted single-cell suspensions were submitted to the University of Michigan Advanced Genomics Core for single-cell RNA-sequencing. Single cells were barcoded, and cDNA libraries constructed on the Chromium platform (10X Genomics, Single Cell 3' v2 chemistry). Paired-end 110 base pair reads were sequenced on NovaSeq 6000 (Illumina).

**Single-cell RNA-sequencing preprocessing**. Raw reads were processed, deconvoluted, droplet filtered, and aligned at the gene level with the Cell Ranger pipeline using default settings (v4.0.0, 10X Genomics) based on the GRCh38 GENCO-DEv32/Ensembl 98 reference transcriptome with STAR v2.5.1b[89]. Previously published single-cell RNA-sequencing raw data of healthy, term placental villous tissue samples from the Database of Genotypes and Phenotypes (Pique-Regi et al., accession number phs001886.v1.p1) SRR10166478 (Sample 3), SRR10166481 (Sample 4), and SRR10166484 (Sample 5)[17] and from the European Genome-Phenome Archive (Tsang et al., accession number EGAS00001002449) (Samples 6–9)[16] were processed identically. The freemuxlet program in the latest version (accessed December 5, 2021) of the "popscle" package was used to assign fetal or maternal origin and identify 736 mosaic doublets for removal based on single nucleotide polymorphisms with minor allele frequency greater than 10% from the 1000 Genomes Phase 3 reference panel (released May 2, 2013)[90]. Per cell quality control criteria were calculated using the quickQCPerCell() function (scater R package, version 1.18.6) with default settings[91] (Supplementary Figs. 2 and 3) and included total unique RNA transcripts (also called unique molecular identifiers), unique genes, and percentage of reads mapping to mitochondrial genes[92]. According to the current recommended best practice, each batch was quality-controlled separately[93]. We excluded 6497 low-quality outlier cells defined as cells with less than 500 unique RNA molecules, less than 200 unique genes, or that were outliers in mitochondrial gene mapping rate. Mitochondrial mapping outliers exceeded four median absolute deviations in samples 1 and 2 (mitochondrial reads >9.2%) or three median absolute deviations in samples 3, 4, and 5 (mitochondrial reads >8.9%) and samples 6, 7, 8C, 8P, 9C, and 9P (mitochondrial reads >9.1%). To generate normalized gene expression data for visualizations and analyses that required normalization, single-cell gene counts were library size normalized by dividing the number of counts by the total number of counts expressed in that cell, multiplied by a scale factor of 10,000, and log-transformed with the Normal-izeData() function (Seurat R package, version 4.1.1).

**Single-cell RNA-sequencing clustering and cluster annotation**. Maternal and fetal cells were split into separate datasets for clustering. To integrate data from cells across study sources and visualize clustering results with uniform manifold projection[34], we used the mutual nearest neighbor batch correction approach via FastMNN from "SeuratWrapper" with default settings (R package, version 0.3.0)[35]. Supervised iterative clustering and sub-clustering with "Seurat" (R package, version 4.0.1) function FindClusters at different resolution parameters were evaluated using cluster stability via clustering trees in "clustree"[94,95]. A priori canonical cell type marker gene expression patterns and cluster marker genes were used to assign cell types to cell clusters (see results). Cells that fell outside cell type clusters or outlying in doublet density calculated with computeDoubletDensity were removed as putative doublets and doublet clusters were identified with findDoubletClusters for removal in "scDblFinder" (R package, version 1.4.0)[96]. 723 maternal-maternal or fetal-fetal putative doublets were excluded after integration and clustering. Using the manually annotated Michigan (this study) and Pique-Regi (phs001886.v1.p1) cell cluster labels as the reference data, Tsang sample (EGAS00001002449) cells were algorithmically annotated with "SingleR" (R package, version 1.6.1)[97] with default settings, followed by manual review. Cells with low prediction certainty (assignment score lower than three median absolute deviations of all cells assigned) were excluded as putative maternal-maternal or fetal-fetal doublets. Fetal sex in Michigan (this study) samples was determined with average normalized *XIST* expression; fetal sex in Pique-Regi and Tsang samples was determined by annotation and confirmed with average normalized *XIST* expression (Supplementary Fig. 4). The final analytic sample included 40,494 cells and 36,601 genes across nine biological replicates, two of which had a technical replicate (Samples 1 and 2) and another two included peripheral subsampling (Samples 8 and 9).

**Single-cell RNA-sequencing differential expression and biological pathway enrichment statistical analysis**. Technical correlation was assessed by Spearman correlation after averaging the normalized expression for each gene by cluster and by technical replicate. Cluster marker genes were identified in "Seurat" with the FindAllMarkers function with default settings on single-cell gene expression counts[92,95]. Specifically, including both maternal and fetal cell types, the expression level in each cell type cluster was compared against the average expression of that gene across all other cell types using the two-tailed Wilcoxon rank sum test with significance defined at a false discovery rate-adjusted *p* value less than 0.05 and a log2 FC cutoff of 0.25. Pairwise cluster markers were identified in "Seurat" with the FindMarkers function with an identical testing regime. Overexpressed genes were ranked by decreasing log2 FC for functional enrichment analysis with "gprofiler2" (R package, version 0.2.0, database version e102_eg49_p15_7a9b4d6) using annotated genes as the universe, excluding electronically generated annotations, and with the g:SCS multiple testing correction method applying a significance threshold 0.05[98].

**In silico testing of deconvolution performance**. To test the performance and robustness of our placental single-cell RNA-sequencing deconvolution reference, we randomly split our analytic single-cell RNA-sequencing dataset into 50% training and 50% testing subsets with balanced cell type proportions[49]. We applied the test subset with the CIBERSORTx Docker container (accessed December 7, 2021) to create a signature gene expression matrix to test deconvolution performance with default settings[59]. To evaluate the reference's robustness to fetal sex and ability to discriminate immune cell types of fetal versus maternal origin, we generated in silico pseudo-bulk test mixtures with known distributions of fetal and maternal cells, as well as male and female placental cells. Test mixtures included all of the 50% testing data, only fetal cells from the test data, only maternal cells from the test data, only female fetal cells from the test data, or only male cells from the test data. For the female and male fetal cell test mixtures, the baseline distribution of maternal cells was maintained by randomly down-sampling the maternal cells and randomly down-sampling the male fetal cells to the number of female fetal cells. We used the signature matrix generated from the training data to estimate constituent cell type proportions in these test mixtures using CIBERSORTx with cross-platform S-mode batch correction and 50 permutations to evaluate imputation goodness-of-fit. Pearson correlations and root mean square error between the test set predicted and actual cell type proportions in the test mixtures were used to assess deconvolution performance.

**Fluorescence-activated cell sorting of major placental cell types from villous tissue**. Villous tissue single-cell suspensions were quickly thawed and stained according to manufacturer's instructions with five fluorescently labeled antibodies (CD9-FITC, CD45-APC, HLA-A,B,C-PE/Cy7, CD31-BV421, and HLA-G-PE) as well as LIVE/DEAD Near-IR stain (Invitrogen) to isolate six viable populations of placental cells by FACS at the University of Michigan Flow Cytometry Core Facility. Initial flow cytometry experiments included fluorescence minus one, single-color compensation, and isotype controls. Isotype controls were found to be the most conservative and were consequently included in all sorting experiments, as well as single-color compensation controls due to the large number of colors used in sorting. The six populations of cells were Hofbauer cells, endothelial cells, fibroblasts, leukocytes, extravillous trophoblasts, and cytotrophoblasts. We developed a five-marker cell surface FACS scheme to sort cytotrophoblasts (HLA-A,B,C-), endothelial cells (CD31+), extravillous trophoblasts (HLA-G+), fibroblasts (CD9+), Hofbauer cells (CD9-), and leukocytes (CD45+/CD9+) from villous tissue (Supplementary Fig. 10)[11,12,14,37,100–106]. Syncytiotrophoblast fragments were enriched from villous tissue digests. We isolated cell type fractions and whole villous tissue from four healthy term, uncomplicated Cesarean sections, labeled Sorted 1 (same sample source as single-cell RNA-sequencing sample 1), Sorted 2, Sorted 3, and Sorted 4. We subjected 24 cell type fractions with sufficient RNA content to RNA-sequencing, including two cytotrophoblast, one endothelial, three extravillous trophoblast, three fibroblast, four Hofbauer cell, four leukocyte, and two syncytiotrophoblast fractions, and five whole tissue samples (Supplementary Table 2).

Detailed antibody information: FITC, marker CD9: Mouse IgG1-kappa, clone HI9a (2.5 µg/mL), BioLegend #312103, lot B188319, BioLegend #312104, lot B232916; isotype control: clone MOPC-21 BioLegend #400107, Lot B199152 (2.5 µg/mL). APC, marker CD45: Mouse IgG1-kappa, clone 2D1, BioLegend #368511, Lot B215062 (0.125 µg/mL); isotype control: clone MOPC-21, BioLegend #400121, lot B216780 (0.125 µg/mL). PE/CY-7, marker HLA-ABC: Mouse IgG2a-kappa, clone W6/32, BioLegend #311429, lot B188649, BioLegend #3111430, lot B238602 (0.44 µg/mL); isotype control: clone MOPC-173, BioLegend #400231, lot B209000 (0.44 µg/mL);. BV421, marker CD31: Mouse IgG1-kappa, clone WM59, BioLegend #303123, lot B204347, BioLegend #303124, lot B232010 (0.625 µg/mL); isotype control: clone MOPC-21, BioLegend #400157, lot B225357 (0.625 µg/mL). PE, marker HLA-G: Mouse IgG2a-kappa, clone 87G, BioLegend #335905, lot B222326, BioLegend #335906, lot B199294 (5 µg/mL); isotype control clone MOPC-173, BioLegend #400211, lot B227641 (5 µg/mL). Mouse IgG1-kappa, clone MEM-G/9, Abcam #24384 Lot GR3176304-1 (2.5 µg/mL); isotype control: monoclonal, Abcam #ab81200, lot GR267131-1 (2.5 µg/mL). Validation information available on the manufacturer's website under the catalog ID for each antibody.

A cutoff of 0.1% events was used to set a series of gates. Cells were first gated on size and granularity (FSC-HxSSC-H) to eliminate debris, followed by doublet discrimination (FSC-HxFSC-W and SSC-HxSSC-W). Ax750 was used to sort on

viability. Extravillous trophoblasts were isolated based on Human Leukocyte Antigen-G (HLA-G) expression (Supplementary Fig. 10a). Cytotrophoblasts are HLA-ABC negative (Supplementary Fig. 10b). HLA-ABC-positive cells were then subjected to a CD45/CD9 gate to isolate Hofbauer cells and a heterogeneous population of leukocytes (Supplementary Fig. 10c). Finally, CD45-/CD9- population is sorted into the endothelial or fibroblast bins based on CD31 expression (Supplementary Fig. 10d).

**Bulk placental tissue and sorted placental cell type RNA extraction and sequencing.** Approximately 2 mg of bulk RNALater-stabilized (Qiagen) bulk villous tissue was added to 350 μL 1% β-mercaptoethanol (Sigma-Aldrich) RLT Buffer Plus (Qiagen) to Lysing Matrix D vials (MP Biomedicals). Samples were disrupted and homogenized on the MP-24 FastPrep homogenizer (MP Biomedicals) at 6 m/s, setting MP24x2 for 35 s. For the homogenized bulk villous tissue, syncytiotrophoblast-enriched fraction, and sorted cell types, RNA extraction was completed according to the manufacturer's instructions using the AllPrep DNA/RNA Mini Kit (Qiagen) and stored at −80 °C. RNA samples were submitted to the University of Michigan Advanced Genomics Core for RNA-sequencing. Ribosomal RNAs were depleted with RiboGone (Takara) and libraries were prepared with the SMARTer Stranded RNA-Seq v2 kit (Takara). Paired- or single-end 50 base pair reads were sequenced on the HiSeq platform (Illumina). Raw RNA reads were assessed for sequencing quality using "FastQC" v0.11.5[107] and "MultiQC" v1.7[108]. Reads were aligned to the GRCh38.p12/ GENCODEv28 reference transcriptome using "STAR" v2.6.0c with default settings[89]. featureCounts from "subread" v1.6.1 was used to quantify and summarize gene expression with default settings[109].

**Sorted placental cell type differential expression analysis and comparison to single-cell results.** For visualizations or analyses that required normalized gene counts, sorted cell type gene counts were library size normalized with the median ratio method using the counts() function (DESeq2 R package, version 1.32.0). As recommended[50], we excluded genes that were not present in at least three samples and did not have an expression of 10 library size-normalized counts. To visualize broad cell type-specific gene expression patterns, we used "DESeq2's" (R package, version 1.32.0) plotPCA() function with the regularized logarithm transformation, blinded to experimental design. Upregulated genes in each cell type were identified using the negative binomial linear model two-tailed Wald test in "DESeq2" (R package, version 1.32.0) adjusted for biological replicate using default settings with contrasts comparing the expression of a gene in one cell type against the average expression across all other cell types at a false discovery rate-adjusted $p$ value less than 0.05 and a $\log_2$ FC cutoff of 1.2[50]. Overexpressed genes were ranked by decreasing log-FC for functional enrichment analysis with "gprofiler2" (R package, version 0.2.0, database version e102_eg49_p15_7a9b4d6) using annotated genes as the universe, excluding electronically generated annotations, and with the default g:SCS multiple testing correction method applying significance threshold adjusted $p$ value of 0.05[98]. To compare sorted and single-cell results, we tabulated unique overlapping differentially expressed genes and overrepresented pathways by cell type (Supplementary Table 3). Peripheral fetal and maternal immune cell types from the single-cell RNA-sequencing data were collapsed to one leukocyte category, cytotrophoblast subtypes to one cytotrophoblast category, and mesenchymal stem cells and fibroblasts to one fibroblast category for this comparison.

We used the CIBERSORTx Docker container (accessed December 7, 2021) to create a signature gene expression matrix for deconvolution from the counts of the single-cell RNA-sequencing data with the following default parameters: differential expression $q$ value < 0.01, no minimum gene expression cutoff, and a 300 gene feature selection floor and a 500 gene feature selection ceiling[99]. We used the signature matrix to estimate constituent cell type proportions in the 4 whole tissue (with 1 additional technical replicate) and 19 sorted or enriched cell type fractions using CIBERSORTx with cross-platform S-mode batch correction and 50 permutations to evaluate imputation goodness-of-fit. We collapsed the high-resolution single-cell cell type cluster labels to the seven cell type fractions we targeted for comparison with sorted cell type results.

**Application: bulk placenta gene expression dataset and CIBERSORTx deconvolution.** Bulk placental tissue microarray gene expression (previously batch-corrected and normalized) from eight preeclampsia case–control studies was downloaded from the NCBI Gene Expression Omnibus (accession number GSE75010) for deconvolution[33]. We used the CIBERSORTx Docker container (accessed December 7, 2021) to create a signature gene expression matrix for deconvolution from the counts of the single-cell RNA-sequencing data with the following default parameters: differential expression $q$ value < 0.01, no minimum gene expression cutoff, and a 300 gene feature selection floor and a 500 gene feature selection ceiling[99]. We used the signature matrix to estimate constituent cell type proportions in GSE75010 using CIBERSORTx with cross-platform S-mode batch correction and 50 permutations to evaluate imputation goodness-of-fit.

**Application: preeclampsia case–control differential cell type abundance, differential gene expression statistical analysis, and mediation analysis.** To test for differences in estimated cell type proportions between preeclampsia cases and controls, estimated cell type proportions for GSE75010 were regressed on

preeclampsia case–control status using beta regression models adjusted for gestational age, sex, and study source[110] (Supplementary Data 8). Statistical significance was assessed using the two-tailed Wald test applying a nominal significance threshold of 0.05. Cell types imputed at zero percent abundance across all samples were excluded. For modelling purposes, zero percent abundance estimates were transformed to $\frac{1}{2}/n$ where $n$ is the number of observations ($n = 330$).

Differential expression analysis was conducted with limma[111] with default linear models adjusted for gestational age, fetal sex, and study source with empirical Bayes standard error moderated $t$-test statistics. A cell type-adjusted model was built on the base model adjusted for gestational age, fetal sex, and study source and additionally adjusted for the first five PCs of deconvoluted cell type proportions (Supplementary Data 9). Statistical significance was assessed at false discovery rate-adjusted $q$ value < 0.05 and a $\log_2$ FC cutoff of 0.1. Differentially expressed genes were descending-ranked by value of the moderated test statistic for gene set enrichment analysis in desktop version GSEA 4.1.0 with the GSEAPreranked tool with default settings against the c5.go.bp.v7.5.1.symbols.gmt gene set database[112,113] (Supplementary Data 10). PC analysis was performed with prcomp() from "stats-package" (R, version 4.0.5) without scaling and with default settings.

A unified mediation and interaction analysis[114] was conducted in "CMAverse" (R package, version 0.1.0)[115] via the g-formula approach[116] to estimate causal randomized-intervention analogs of natural direct and indirect effects[117] through direct counterfactual imputation. The model was operationalized with preeclampsia status as the binary exposure, $\log_2$ transformed gene expression intensity as the continuous outcome, and the first five PCs of deconvoluted cell type proportions as continuous mediators. Baseline covariates included fetal sex and study source. Continuous gestational age was included as a confounder of the mediator-outcome relationship affected by the exposure. Confidence intervals were bootstrapped with 1000 boots with otherwise default settings. Statistical tests were two-tailed and interpreted at a $p$ value significance threshold of 0.05.

**Statistics and reproducibility.** Technical replication measured by average intra-cluster gene expression between technical replicates was tested via the two-tailed Spearman correlation test within Samples 1 and 2 assessed across all 32,738 common genes. The number of cells contributing expression data for each cell type is available in Table 1. Single-cell cluster marker genes were identified in "Seurat" with the FindAllMarkers function with default settings on single-cell gene expression counts[92,95]. Specifically, including cells from both maternal and fetal cell types, the expression level in each cell type cluster was compared against the average expression of that gene across all other cell types using the two-tailed Wilcoxon rank sum test with significance defined at a false discovery rate-adjusted $p$ value less than 0.05 and a $\log_2$ FC cutoff of 0.25 ($n = 40,494$ cells). The final analytic sample included 40,494 cells and 36,601 genes across nine biological replicates, two of which had a technical replicate (Samples 1B and 2B) and another two included peripheral subsampling (Samples 8P and 9P). Pairwise cluster markers were identified in "Seurat" with the FindMarkers function with an identical testing regime ($n = 6132$ cells for proliferative vs. non-proliferative cytotrophoblasts). Overexpressed genes were ranked by decreasing log-FC for functional enrichment analysis with "gprofiler2" (R package, version 0.2.0, database version e102_eg49_p15_7a9b4d6) using annotated genes as the universe, excluding electronically generated annotations, and with the default g:SCS multiple testing correction method applying significance threshold adjusted $p$ value of 0.05[98]. Overexpressed genes per cell type cluster are available in Supplementary Data 2 and ontology results in Supplementary Data 3. Overexpressed genes and related enrichment results comparing proliferative to non-proliferative cytotrophoblasts are available in Supplementary Data 1.

Upregulated genes in each cell type were identified using the negative binomial linear model two-tailed Wald test in "DESeq2" (R package, version 1.32.0) adjusted for biological replicate using default settings with contrasts comparing the expression of a gene in one cell type against the average expression across all other cell types at a false discovery rate-adjusted $p$ value less than 0.05 and a $\log_2$ FC cutoff of 1.2[50] ($n = 19$ cell type fraction samples with breakdown by cell type available in Supplementary Table 2). Overexpressed genes were ranked by decreasing log-FC for functional enrichment analysis with "gprofiler2" (R package, version 0.2.0, database version e102_eg49_p15_7a9b4d6) using annotated genes as the universe, excluding electronically generated annotations, and with the default g:SCS multiple testing correction method applying significance threshold adjusted $p$ value of 0.05[98]. Differentially expressed genes per cell type available in Supplementary Data 4 and number of differentially expressed genes are summarized in Supplementary Fig. 12. Ontology results are available in Supplementary Data 5.

Bulk placental tissue microarray gene expression (previously batch-corrected and normalized) from eight preeclampsia case–control studies was downloaded from the NCBI Gene Expression Omnibus (GSE75010) for deconvolution ($n = 330$)[33]. We used the CIBERSORTx Docker container (accessed December 7, 2021) to create a signature gene expression matrix for deconvolution from the counts of the single-cell RNA-sequencing data with the following default parameters: differential expression $q$ value < 0.01, no minimum gene expression cutoff, and a 300 gene feature selection floor and a 500 gene feature selection ceiling[99]. We used the signature matrix to estimate constituent cell type proportions in GSE75010 using CIBERSORTx with cross-platform S-mode batch correction and 50 permutations to evaluate imputation goodness-of-fit.

To test for differences in estimated cell type proportions between preeclampsia cases and controls ($n = 330$), estimated cell type proportions for GSE75010 were regressed on preeclampsia case–control status using beta regression models ($n = 25$ cell type proportion outcomes) adjusted for gestational age, sex, and study source[110]. Cell types imputed at zero percent abundance across all samples were excluded ($n = 2$ excluded: fetal naïve CD4+ T cells and fetal GZMB+ natural killer cells). Statistical significance was assessed using the two-tailed Wald test applying a nominal significance threshold of 0.05.

Differential expression analysis was conducted in limma[111] with default settings using linear models ($n = 14,651$ genes) adjusted for gestational age, fetal sex, and study source ($n = 330$). A cell type-adjusted model was built on the base model additionally adjusted for the first five PCs of deconvoluted cell type proportions. PC analysis was performed with prcomp from "stats-package" (R, version 4.0.5) without scaling and default settings. Statistical significance was assessed at false discovery rate-adjusted $q$ value < 0.05 and a $\log_2$ FC cutoff of 0.1. Differentially expressed genes were descending-ranked by the value of the moderated test statistic for gene set enrichment analysis in desktop version GSEA 4.1.0 with the GSEAPreranked tool with default settings against the c5.go.bp.v7.5.1.symbols.gmt gene set database[112,113].

A unified mediation and interaction analysis[114] was conducted in "CMAverse" (R package, version 0.1.0)[115] via the g-formula approach[116] to estimate causal randomized-intervention analogs of natural direct and indirect effects[117] through direct counterfactual imputation. The model ($n = 330$) was operationalized with preeclampsia status as the binary exposure, normalized $\log_2$ gene expression signal intensity as the outcome, and the first five PCs of deconvoluted cell type proportions as continuous mediators. Baseline covariates included fetal sex and categorical study source. Continuous gestational age was included as a confounder of the mediator-outcome relationship affected by the exposure. Confidence intervals were bootstrapped with 1000 boots with otherwise default settings. Statistical tests were two-tailed and interpreted at a $p$ value significance threshold of 0.05.

**Reporting summary**. Further information on research design is available in the Nature Portfolio Reporting Summary linked to this article.

## Data availability

Raw placental single-cell RNA-sequencing and raw placental bulk RNA-sequencing generated by this study are freely available in the Gene Expression Omnibus repository (accession number GSE182381). The cell type signature matrix and related files to deconvolute bulk gene expression measures are available through the Gene Expression Omnibus (accession number GSE182381) as supplementary material for download. This study uses data generated by The Chinese University of Hong Kong (CUHK) Circulating Nucleic Acids Research Group, as reported by Tsang et al. in *Proc. Natl Acad. Sci. USA* (doi: 10.1073/pnas.1710470114, accession number EGAS00001002449)[16,88]. The placental single-cell RNA-sequencing data that support the findings of this study can be accessed through the Database of Genotypes and Phenotypes (accession number phs001886.v1.p1)[17,87] and the European Genome-Phenome Archive (accession number EGAS00001002449)[16,88]. The preeclampsia case–control microarray data that support the findings of this study are available in Gene Expression Omnibus repository (accession number GSE75010)[33,118]. Source data underlying Figs. 3–5 are presented in Supplementary Data 8–10, respectively.

## Code availability

All scripts to perform preprocessing, analyses, and deconvolution are available[119].

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

## Acknowledgements

This research was supported by the Michigan Lifestage Environmental Exposures and Disease center (P30 ES017885), Michigan State University's Environmental Influences on Child Health Outcomes program (UG3 OD023285, UH3 OD023285), the University of Michigan M-cubed pilot grant program, the Puerto Rico Testsite for Exploring Contamination Threats (P42 ES017198), and the NIH R35 RIVER award (ES031686). K.A.C. was supported by the National Institutes of Health (T32 HG00040). J.A.C. was supported by the Ravitz Family Foundation, the Forbes Institute for Cancer Discovery at the University of Michigan Rogel Cancer Center, and the National Institutes of Health (R01 ES028802, U01 ES026697, and P30 CA046592). M.P. was supported by the National Institutes of Health (T32 ES007062). E.R.E was supported by the National Institutes of Health (T32 DK071212). K.M.B. was supported by research grants from the National Institute of Environmental Health Sciences (R01 ES025531; R01 ES025574). We acknowledge the submitting investigators of previously published placental single-cell RNA-sequencing data used in this study. This study uses data generated by The Chinese University of Hong Kong (CUHK) Circulating Nucleic Acids Research Group, as reported by Tsang et al. in *Proc. Natl Acad. Sci. USA* (doi: 10.1073/pnas.1710470114, accession number EGAS00001002449). We also acknowledge the submitting investigators of Database of Genotypes and Phenotypes (accession number phs001886.v1.p1), whose work was sponsored in part by the Perinatology Research Branch, Division of Obstetrics and Maternal-Fetal Medicine, Division of Intramural Research, Eunice Kennedy Shriver National Institute of Child Health and Human Development, National Institutes of Health, U.S. Department of Health and Human Services.

## Author contributions

K.A.C.: conceptualization, methodology, software, validation, formal analysis, investigation, data curation, writing—original draft, visualization. J.A.C.: conceptualization, methodology, software, validation, investigation, resources, writing—review and editing, visualization, supervision, project administration, funding acquisition. M.P.: investigation, writing—review and editing. J.F.D.: software, validation, data curation. E.R.E.: writing—review and editing. S.S.H.: methodology, writing—review and editing, funding acquisition. S.E.D.: methodology, writing—review and editing. D.C.D.: writing—review and editing, funding acquisition. J.M.G.: writing—review and editing, funding acquisition. R.L.-C.: conceptualization, methodology, resources, writing—review and editing, supervision, project administration, funding acquisition. V.P.: conceptualization, resources, writing—review and editing, project administration, resources, funding acquisition. K.M.B.: conceptualization, methodology, software, validation, formal analysis, investigation, resources, data curation, writing—original draft, visualization, supervision, project administration, funding acquisition.

## Competing interests

The authors declare no competing interests.
