## [Peer Review File · Communications Biology]

Reviewers' comments:

Reviewer #1 (Remarks to the Author):

The manuscript by Campbell et al. entitled "Placental gene expression-based cell type deconvolution: Cell proportions drive preeclampsia gene expression differences" explores using a reference-based deconvolution approach for placental gene-expression using data derived from single-cell RNAseq (scRNAseq). The authors applied the approach in 330 publicly available samples to test associations with preeclampsia. The authors identified 19 fetal and eight maternal cell types in their reference and found that no genes were differentially expressed after correcting for cell types. The idea is enticing, and this work is trying to fill a critical gap in reproductive epidemiology. There are a few areas that are not clear and may require some expansion.

Comments:

Page 5, Results, supplementary table 1: can you elaborate on the cell types in the sc-RNAseq? What are the composite cells?

Supplementary Figure 2: There is a noticeable batch effect between the two sources of the data. How were those differences accounted for? Are any specific concerns about the quality of the Pique-Regi dataset?

Figure 1: some of the labels are unclear about the cell type assigned. What are the CD8+ Cytotoxic T cells (maternal and fetal)? Are those memory cells? What about the naive CD8+ T cells?

Supplementary figure 6, supplementary table 4, table 1: The technical effects are more problematic here. The two replicates are mostly immune cells, while the different trophoblast cells are only captured in the Pique-Regi dataset. Could you elaborate on why those differences appeared? Explain to the reader the potential limitations and, if you can, discuss why these cells were not captured in your replicate dataset. One necessary clarification for the reader is required on why there is a discordance between your scRNAseq and the FACS results for sample 1.

Page 6, Supplementary Figure 8: How was the selection of the transcripts performed for this analysis? Are all the transcripts included in this analysis? Did you mask low counts or selected per variability first? Would you mind justifying and adding to the text as needed?

Supplementary Figure 9: What is the scale of the normalized counts? Can you explain a bit about the differences between abundances when analyzing bulk and scRNAseq, and how can that affect those relations? What happened when you applied your reference to the sorted data? Were the cells concordant or not?

Signature matrix text and supplementary figure 10: The authors mention trying to deconvolve 27 fetal and maternal cell types. I need some clarification here as I am lost on this step. The authors used genetic variation to infer fetal vs. maternal for the single-cell experiment, which makes sense in that context; I do not see how you can transfer that to the bulk RNAseq deconvolution using transcripts. My impression looking at your heatmap is that your markers between maternal and fetal are minimal, and this is only generating noise. Would you please elaborate in your answer on why separate those and how you can ascertain that the process works for your bulk samples? This point is essential to understand your results on supplementary figure 11, which are primarily zeroes with some signals affecting the specific cell types in your other analyses.

Single-cell enrichment: According to Yabe et al., the strainers should be in the range of 40-70 micrometers. Could you explain why the modification to 100 is in your protocol? This is important for the future reproducibility of your protocol.

Reviewer #2 (Remarks to the Author):

Submitted report:

In this article, Campbell et al. generate single-cell RNA-seq data in term placenta and integrate it with existing term placenta single-cell RNA-seq data. The goal was to create a deconvolution reference of fetal and maternal placenta cell types, which could be used as a resource to those using bulk sequencing. As an application of the resource, the authors use their reference to deconvolute microarray data generated in preeclamptic (PE) and control placenta. They report that bulk gene expression differences could be due to differences in cell type composition in PE versus control samples. The deconvolution resource is a clever and interesting idea, however there are issues with the samples and analyses that made me question the deconvoluted data and robustness of conclusions that can be made by using the resource. Specific comments are below:

-Table 1 - Cell type percentages from different studies are very different, and this is concerning. Samples 1 and 2 have a lot of fetal B cells and other immune cells, but not a lot of cytotrophoblast, no EVT. SCT are only really present in sample 4. This is likely due to sampling as well as technique in preparing samples. These differences would surely impact the deconvolution reference, as well as conclusions from other samples, depending on how sampling was done.

-There are additional published term placenta single-cell RNA-seq data - how come they were not integrated?

-Supplementary table 3 - the main text gives p-adj but the supplemental table only has p-value. The ontology terms that are specific to a particular cell type should be visualized in some way - cell type defining processes are highlighted for SCT, but not other cell types. Do ontologies for other cell types make sense? Also for supplemental excel file it would be helpful to have ontology results for each cell type on a different tab.

-Supplementary table 4 that was downloaded seems to be different from what is referred to in the text and given in the supplementary figures file.

-It is concerning how similar the different cell types appear in figure S8 - does this indicate they were not sorted properly? One of the leukocyte samples appears to be an outlier. How would removing it impact the PCA?

-Presumably, cell-type defining transcripts, as determined by DESeq2 analysis, should be expressed at a higher fold in the cell type being compared to all other cell types. However, it does not appear that a fold-change cutoff was implemented according to the methods. Looking at the numbers reported in figure 2 and supplementary table 5 though, it does seem like a fold of 2 might have been used. For some cell types in table S5, the number of upregulated genes is not the same as what is depicted in figure 2.

-Supplementary Table S6 - how come SCT are not enriched for hormone related terms, as were enriched for the single-cell data? Similar to the single cell data, the ontology terms that are specific to a particular cell type should be visualized in some way - cell type defining processes are highlighted for SCT, but not other cell types. Also for supplemental excel file it would be helpful to have ontology results for each cell type on a different tab.

-Both Pearson and Spearman correlations are provided for single-cell vs bulk expression. Pearson correlations are very low for all cell types, and spearman is low for some of the cell types. To help convince there is a correlation, correlations between different cell types can be reported, which would

presumably show that cross-cell type correlations are much lower than same cell type correlations. Are the genes considered as marker genes according to single cell data similar to those that are upregulated in the corresponding cell type in bulk data?

-I am confused about the signature gene expression matrix used for deconvolution. According to figure S10, most of the genes have low expression. How would these help deconvolute data? A scalebar for expression should be provided in the figure, as well as an explanation for what is shown in the plot above the heatmap.

-Related to the above, no minimum gene expression cutoff is used in CIBERSORTx analysis, but wouldn't this, as well as a fold cutoff provide more robust gene signatures?

-The authors indicate that using the the cell type naive base model 1,224 genes are differentially expressed. However, this is not using a fold change, which is appropriate to use in this analysis. It looks like using fold-change would bring the number of differentially expressed genes to a very small number, making the cell type-adjusted results not too different.

-It appears that even if a fold-change cutoff were to be used, FLT1 and FSTL3, both which have been associated PE (the authors do note this and do further analysis with FLT1), would have been identified in the standard differential expression analysis, but not in the model that considers cell type. Does using deconvolution then lead to us losing information about differentially expressed genes in PE? Or is the point to use both methods to look at genes and then possible cell types that are impacted?

-It would be helpful to know what numbers were used for filtering criteria in scRNAseq (rather than just exceeding four median absolute deviations). How come 4 deviations were used for samples 1 and 2 and 3 median deviations were used for samples 3-5?

-Should male and female be separated for the purpose of deconvolution? There are 2 biological female and 3 biological male studies.

-Supplementary table 2 would be more useful if there was a separate tab for each cell type, and each tab had only the significantly upregulated markers for that cell type.

-Supplementary table 5 - it would be helpful if results for each cell type was in a different tab

-The authors provide a github page for the lab, but there is no specific information about this study on that page. Also, the specific link to this study should be provided in the paper, rather than the lab page.

We thank the editor for the opportunity to resubmit this manuscript. The reviewers provided incredibly thorough feedback, and while they highlighted the importance of this research, they identified important areas for improvement of the research. In particular, the reviewers suggested adding additional datasets to the placenta cell type reference panel. In response to reviewer feedback, we have successfully incorporated an additional dataset into our manuscript and improved the rigor of multiple aspects of the paper. Our specific changes are described point by point below.

Reviewers' comments:

Reviewer #1 (Remarks to the Author):

Comment 1: The manuscript by Campbell et al. entitled "Placental gene expression-based cell type deconvolution: Cell proportions drive preeclampsia gene expression differences" explores using a reference-based deconvolution approach for placental gene-expression using data derived from single-cell RNAseq (scRNAseq). The authors applied the approach in 330 publicly available samples to test associations with preeclampsia. The authors identified 19 fetal and eight maternal cell types in their reference and found that no genes were differentially expressed after correcting for cell types. The idea is enticing, and this work is trying to fill a critical gap in reproductive epidemiology. There are a few areas that are not clear and may require some expansion.

Response 1: We thank the reviewer for recognizing the utility of this research for reproductive epidemiology.

Comment 2: Page 5, Results, supplementary table 1: can you elaborate on the cell types in the scRNAseq? What are the composite cells?

Response 2: Yes, single cell RNA analyses allow for the identification of cell clusters. Assigning these clusters as cell types and cell sub types is important for biologic interpretability of the findings. We have added several details to the rationale for and method of assigning cell types from the scRNA-seq results.

We see how using the term "composite" can be vague. To avoid this potential confusion, we have replaced "composite" with "whole tissue" throughout the manuscript.

Revisions made:

Results: "Cell type clustering decisions balanced cluster stability, resolution, and biologic plausibility with prior knowledge. For example, maternal CD14+ and FCGR3A+ monocytes were stable as independent clusters, despite their relatively similar gene expression profiles. If desired, downstream analyses could collapse these subtypes into a single maternal monocyte cluster."

Methods: "...and five whole tissue samples".

Comment 3: Supplementary Figure 2: There is a noticeable batch effect between the two sources of the data. How were those differences accounted for? Are any specific concerns about the quality of the Pique-Regi dataset?

Response 3: Yes, batch effects were expected between the samples sequenced at Michigan and those pulled from the literature. To address these batch effects, we used the mutual nearest-neighbor integration algorithm to integrate datasets across batch [1]. We have clarified usage of this as the batch correction approach in the Methods.

Revisions made:

Methods: “To integrate data from cells across study sources, the mutual nearest neighbor batch correction approach with default settings was used ...”

We applied consistent quality control standards and observed similar single cell data quality metrics across the three studies. The study specific quality metrics are described in Supplementary Figures 2-3.

Comment 4: Figure 1: some of the labels are unclear about the cell type assigned. What are the CD8+ Cytotoxic T cells (maternal and fetal)? Are those memory cells? What about the naive CD8+ T cells?

Response 4: Thank you for the opportunity to improve Figure 1 and clarify. We used the term “CD8+ Cytotoxic T cells” to refer to activated T cells, which would include CD8+ Memory T cells. We relabeled this cell type as “CD8+ Activated T cells” because we agree “cytotoxic” is not a clear label to identify non-naïve CD8+ T cells. We use “Naïve CD8+ T cells” to describe CD8+ T cells that are distinguished from CD8+ Activated T cells by their expression of *CCR7* and *IL7R*. To assign maternal and fetal labels, genetic diversity in the transcripts was analyzed with the freemuxlet program.

Comment 5: Supplementary figure 6, supplementary table 4, table 1: The technical effects are more problematic here. The two replicates are mostly immune cells, while the different trophoblast cells are only captured in the Pique-Regi dataset. Could you elaborate on why those differences appeared? Explain to the reader the potential limitations and, if you can, discuss why these cells were not captured in your replicate dataset. One necessary clarification for the reader is required on why there is a discordance between your scRNAseq and the FACS results for sample 1.

Response 5: Yes, we did observe differences in the numbers and relative proportion of cell types captured across study sources. To address this and other comments, we have now added a third term placenta single-cell RNA-seq dataset [2] to our paper, which increases the rigor and generalizability of our findings.

Each of the three source studies (our Michigan sample, the Pique-Regi sample, and the new Tsang sample) used different placental tissue dissociation and cell enrichment methods. Cell types were likely differentially susceptible to the experimental conditions in each study. Our inclusion of multiple datasets ensures that the widest array of placental cell types are represented in the reference panel, and is a strength of our approach. For example, large and multinucleated syncytiotrophoblasts may be vulnerable to dissociation approaches, and we expected these cells to be captured at lower rates in longer dissociations and/or a freeze-thaw cycle. However, endothelial cells may require more rigorous dissociation to separate from the basement membrane and be more successfully captured in different conditions. We have augmented our statements in the paper to specifically define dissociation bias and address why Michigan samples 1 and 2, specifically, had low placental cell counts.

Revisions made:

Discussion: “The low representation of some cell types such as trophoblasts in our single-cell RNA-sequencing results from the Michigan study suggest that these cell types may be especially sensitive to dissociation and disintegrate before transcript capture, commonly referred to as dissociation bias [3]. Michigan samples 1 and 2 also included a cryopreservation step like those employed in large-scale epidemiological studies that may have exacerbated dissociation bias [4]; this applies to both single-cell and sorted cell type experiments.”

In our first submission, we used the reference panel to deconvolute the bulk RNA samples. We were intrigued by the reviewer idea to also deconvolute the sorted cell RNA samples. This analyses and

these findings are now included in the manuscript, including a new Supplemental Table, which is now Supplementary Table 7.

Methods: “We used the CIBERSORTx Docker container (accessed 2021-12-07) to create a signature gene expression matrix for deconvolution from the counts of the single-cell RNA-sequencing data with the following default parameters: differential expression q-value<0.01, no minimum gene expression cutoff, and a 300 gene feature selection floor and a 500 gene feature selection ceiling [5]. We used the signature matrix to estimate constituent cell type proportions in the 4 whole tissue (with 1 additional technical replicate) and 19 sorted or enriched cell type fractions using CIBERSORTx with cross-platform S-mode batch correction and 50 permutations to evaluate imputation goodness-of-fit. We applied the deconvolution reference to estimate cell proportions in the 4 whole tissue (with 1 additional technical replicate) and 19 sorted or enriched cell type fractions. We collapsed the high-resolution single-cell cell type cluster labels to the seven cell type fractions we targeted for comparison with sorted cell type results.”

Results: “We applied the deconvolution reference to estimate cell proportions in the 4 whole tissue (with 1 additional technical replicate) and 19 sorted or enriched cell type fractions. We collapsed the high-resolution single-cell cell type cluster labels to the seven cell type fractions we targeted for isolation downstream analyses (Supplementary Table 7 – Sheet 1). All deconvoluted samples exhibited high goodness-of-fit between original bulk mixtures and the estimated cell type proportion mixtures (p-values<0.001). Among the signature genes, original bulk and estimated cell type fractions had a Pearson correlation (mean \pm standard deviation) of 0.73 ± 0.11 and root mean square error of 0.88 ± 0.04 (Supplementary Table 7 – Sheet 2). Deconvolution results (mean \pm standard deviation) suggest we successfully isolated fibroblast- (n=3, $74.7\% \pm 0.6\%$) and leukocyte-enriched (n=4, $82.3\% \pm 24.8\%$) cell type fractions. Other cell type targets were less successful (range 0%-26% estimated purity). The Hofbauer cell fraction was predicted to be mostly leukocytes (n=4, $91.5\% \pm 0.5\%$) and not specifically the resident placental macrophage that was successfully identified in the single-cell RNA-sequencing dataset.

Comment 6: Page 6, Supplementary Figure 8: How was the selection of the transcripts performed for this analysis? Are all the transcripts included in this analysis? Did you mask low counts or selected per variability first? Would you mind justifying and adding to the text as needed?

Response 6: We agree that including these technical details will be important for reproducing the research. We added these details to the methods and results sections. They are also provided in our code on GitHub.

Revisions made:

Methods: “As recommended [6], for all downstream analyses, we excluded genes from the differential expression analysis that were not present in at least three samples and did not have an expression of 10 library size-normalized counts. To visualize broad cell type-specific gene expression patterns, we used ‘DESeq2’s (R package, version 1.32.0) plotPCA() function with the regularized logarithm transformation, blinded to experimental design.”

Results: “As recommended [6], we excluded 19,048 genes that were not present in at least 3 samples and an additional 865 genes that did not have a cumulative library size-normalized count of at least 10. Principal components analysis of whole-transcriptome sorted-cell bulk RNA sequencing normalized counts revealed three loosely defined clusters along principal component (PC) 1: (i) whole tissue and syncytiotrophoblast, a distributed cluster of fibroblasts, cytotrophoblasts, extravillous

trophoblasts, and endothelial cells along PC2 (ii), and (iii) a tighter cluster of Hofbauer cells and leukocytes (Supplementary Figure 10).”

Comment 7: Supplementary Figure 9: What is the scale of the normalized counts? Can you explain a bit about the differences between abundances when analyzing bulk and scRNAseq, and how can that affect those relations? What happened when you applied your reference to the sorted data? Were the cells concordant or not?

Response 7: To aid interpretation, we included more detail in the methods on the scales of bulk and single cell RNA sequencing data. Due to various manuscript changes, this is now Supplementary Figure 12.

Revisions made:

Methods: “To aggregate sorted cell type bulk RNA-sequencing gene expression, sorted cell type gene counts were library-size normalized with the median ratio method using the counts() function (DESeq2 R package, version 1.32.0), log-transformed, and averaged across all libraries of that cell type. To aggregate single-cell RNA-sequencing normalized expression by cell type, single-cell counts were library-size normalized by dividing the number of counts by the total number of counts expressed in that cell, multiplied by a scale factor of 10,000, log-transformed, and averaged across all cells of that cell type with the NormalizeData() and AverageExpression() functions (Seurat R package, version 4.1.1).”

We have added additional context in the results section to comparing bulk and single-cell RNA sequencing results.

Results: “However, given their major technical differences, there is no method to account for batch effects between single-cell and sorted bulk RNA sequencing results.”

In our first submission, we used the reference panel to deconvolute the bulk RNA samples. We were intrigued by the reviewer idea to also deconvolute the sorted cell RNA samples. This analyses and these findings are now included in the manuscript, including a new Supplementary Table, which his now Supplementary Table 7.

Methods: “We used the CIBERSORTx Docker container (accessed 2021-12-07) to create a signature gene expression matrix for deconvolution from the counts of the single-cell RNA-sequencing data with the following default parameters: differential expression q-value<0.01, no minimum gene expression cutoff, and a 300 gene feature selection floor and a 500 gene feature selection ceiling [5]. We used the signature matrix to estimate constituent cell type proportions in the 4 whole tissue (with 1 additional technical replicate) and 19 sorted or enriched cell type fractions using CIBERSORTx with cross-platform S-mode batch correction and 50 permutations to evaluate imputation goodness-of-fit. We applied the deconvolution reference to estimate cell proportions in the 4 whole tissue (with 1 additional technical replicate) and 19 sorted or enriched cell type fractions. We collapsed the high-resolution single-cell cell type cluster labels to the seven cell type fractions we targeted for comparison with sorted cell type results.”

Results: “We applied the deconvolution reference to estimate cell proportions in the 4 whole tissue (with 1 additional technical replicate) and 19 sorted or enriched cell type fractions. We collapsed the high-resolution single-cell cell type cluster labels to the seven cell type fractions we targeted for isolation downstream analyses (Supplementary Table 7 – Sheet 1). All deconvoluted samples exhibited high goodness-of-fit between original bulk mixtures and the estimated cell type proportion

mixtures (p-values<0.001). Among the signature genes, original bulk and estimated cell type fractions had a Pearson correlation (mean \pm standard deviation) of 0.73 ± 0.11 and root mean square error of 0.88 ± 0.04 (Supplementary Table 7 – Sheet 2). Deconvolution results (mean \pm standard deviation) suggest we successfully isolated fibroblast- (n=3, $74.7\% \pm 0.6\%$) and leukocyte-enriched (n=4, $82.3\% \pm 24.8\%$) cell type fractions. Other cell type targets were less successful (range 0%-26% estimated purity). The Hofbauer cell fraction was predicted to be mostly leukocytes (n=4, $91.5\% \pm 0.5\%$) and not specifically the resident placental macrophage that was successfully identified in the single-cell RNA-sequencing data.”

Comment 8: Signature matrix text and supplementary figure 10: The authors mention trying to deconvolve 27 fetal and maternal cell types. I need some clarification here as I am lost on this step. The authors used genetic variation to infer fetal vs. maternal for the single-cell experiment, which makes sense in that context; I do not see how you can transfer that to the bulk RNAseq deconvolution using transcripts. My impression looking at your heatmap is that your markers between maternal and fetal are minimal, and this is only generating noise. Would you please elaborate in your answer on why separate those and how you can ascertain that the process works for your bulk samples? This point is essential to understand your results on supplementary figure 11, which are primarily zeroes with some signals affecting the specific cell types in your other analyses.

Response 8: To improve the communication of our overall approach and multiple types of analyses, we added a graphical conceptual diagram to the manuscript (Supplementary Figure 1).

We agree that these are relevant points of concern. To address these concerns, we conducted a new *in silico* deconvolution validation analysis that is now a primary figure (Figure 2). We added methods and results text for this new analysis. Our results demonstrate that our deconvolution reference is largely robust to even extremely imbalanced distributions of fetal vs. maternal cell type abundances as well as male vs. female placental cell type abundances. We also communicate the potential limitations of this approach specifically in assigning fetal vs. maternal origin in cell types that are common to fetal and maternal individuals.

Revisions made:

Methods: **“In silico testing of deconvolution performance**

To test the performance and robustness of our placental single-cell RNA sequencing deconvolution reference, we randomly split our analytic single-cell RNA sequencing dataset into 50% training and 50% testing subsets with balanced cell type proportions [36]. We applied the test subset with CIBERSORTx Docker container (accessed 2021-12-07) to create a signature gene expression matrix to test deconvolution performance with default settings [84]. To evaluate the reference’s robustness to fetal sex and ability to discriminate immune cell types of fetal versus maternal origin, we generated *in silico* pseudo-bulk test mixtures with known distributions of fetal and maternal cells, as well as male and female placental cells. Test mixtures included all of the 50% testing data, only fetal cells from the test data, only maternal cells from the test data, only female fetal cells from the test data, or only male cells from the test data. For the female and male fetal cell test mixtures, the baseline distribution of maternal cells was maintained by randomly down-sampling the maternal cells and randomly down-sampling the male fetal cells to the number of female fetal cells. We used the signature matrix generated from the training data to estimate constituent cell type proportions in these test mixtures using CIBERSORTx with cross-platform S-mode batch correction and 50 permutations to evaluate imputation goodness-of-fit. Pearson correlations and root mean square error between the test set

predicted and actual cell type proportions in the test mixtures were used to assess deconvolution performance.”

Results: “**Single-cell RNA sequencing deconvolution reference exhibits excellent in silico performance**”

Based on the single cell data, we created a placental signature gene matrix that incorporated expression information across an algorithmically selected 5,229 signature genes to estimate the cellular composition of 27 fetal and maternal cell types from whole tissue gene expression data (Supplementary Figure 8). To test the performance and robustness of this placental single-cell RNA sequencing deconvolution reference, we randomly split our analytic single-cell RNA sequencing dataset into 50% training and 50% testing subsets with balanced cell type proportions [36]. The same training dataset was used for each comparison; test mixtures were generated from the testing half of the dataset. Using a signature gene expression matrix generated from the training data, we estimated cell type composition in in silico pseudo-bulk testing data mixtures of known cell type composition with varying contributions of fetal vs. maternal origin cells and male vs. female fetal cells (Figure 2). In all mixtures, the 27 predicted and actual cell type proportions were correlated (p -value <0.001 for each test). In the primary deconvolution analysis of all cell types at their natural rates, estimated and actual cell type proportions had a Pearson correlation coefficient of 0.956 (95% CI [0.904, 0.980]). The worst performance was under the unrealistic scenario that the mixture was composed entirely of maternal cell types with a Pearson correlation of 0.743 (95% CI [0.505, 0.876]) between actual estimated cell type proportions. Our deconvolution reference was also robust to fetal sex when only male fetal cells (Pearson correlation = 0.894, 95% CI [0.779, 0.951]) were included, or only female fetal cells (Pearson correlation = 0.983, 95% CI [0.964, 0.993]). Together, these results show that our reference panel can successfully deconvolute placental tissues, though some maternal cell types common to both mother and fetus may be erroneously labelled fetal in the absence of fetal examples of those cell types.”

Comment 9: Single-cell enrichment: According to Yabe et al., the strainers should be in the range of 40-70 micrometers. Could you explain why the modification to 100 is in your protocol? This is important for the future reproducibility of your protocol.

Response 9: We agree that this is an important detail for the protocol’s reproducibility and have clarified the methods section. The Yabe et al. citation specifies that their 40-70 μ m and >70 μ m fraction both contained syncytiotrophoblasts; in addition, the >70 μ m fraction was less contaminated than the 40-70 μ m fraction [7]. We therefore included the >70 μ m fraction since it was largely syncytial but with a 100 μ m ceiling to limit the capture of undigested tissue fragments.

Revisions made:

Methods: “Single-cell suspensions were size-filtered at 100 μ m to remove undigested tissue [8, 9] and subsequently 40 μ m. To collect a syncytiotrophoblast-enriched fraction, the fraction between 40 μ m and 100 μ m was washed from the 40 μ m strainers, adapting a previous protocol that collected syncytiotrophoblasts throughout this size range [7].”

Reviewer #2 (Remarks to the Author):

Submitted report:

Comment 1: In this article, Campbell et al. generate single-cell RNA-seq data in term placenta and integrate it with existing term placenta single-cell RNA-seq data. The goal was to create a deconvolution reference of fetal and maternal placenta cell types, which could be used as a resource to those using bulk sequencing. As an application of the resource, the authors use their reference to deconvolute microarray data generated in preeclamptic (PE) and control placenta. They report that bulk gene expression differences could be due to differences in cell type composition in PE versus control samples. The deconvolution resource is a clever and interesting idea, however there are issues with the samples and analyses that made me question the deconvoluted data and robustness of conclusions that can be made by using the resource. Specific comments are below:

Response 1: We thank the reviewer for their recognition of the importance of this type of study and we appreciate their constructive feedback. Our responses to specific comments follow.

Comment 2: -Table 1 - Cell type percentages from different studies are very different, and this is concerning. Samples 1 and 2 have a lot of fetal B cells and other immune cells, but not a lot of cytotrophoblast, no EVT. SCT are only really present in sample 4. This is likely due to sampling as well as technique in preparing samples. These differences would surely impact the deconvolution reference, as well as conclusions from other samples, depending on how sampling was done.

Response 2: We did observe differences in the counts and relative proportions of cell types captured across biological replicates in our study, as well as differences in cell type proportions captured across studies. This may partially reflect biologic heterogeneity across placentas in cell composition, or differences in cell vulnerability to dissociation procedures. Each of the three source studies (our Michigan sample, the Pique-Regi sample, and the new Tsang sample) used different placental tissue dissociation and cell enrichment methods. Cell types were likely differentially susceptible to the experimental conditions in each study. Our inclusion of multiple datasets ensures that the widest array of placental cell types are represented in the reference panel, and is a strength of our approach. In particular, large, multinucleated syncytiotrophoblasts may be especially sensitive to dissociation approaches, and we expected these cells to be captured at lower rates. However, endothelial cells may require rigorous dissociation to separate from the basement membrane and be more successfully captured in different conditions. We refined our statements in the paper to specifically define dissociation bias and address why Michigan samples 1 and 2, specifically, had low placental cell counts.

Revisions made:

Discussion: "The low representation of some cell types such as trophoblasts in our single-cell RNA-sequencing results suggest that these cell types may be especially sensitive to dissociation and disintegrate before transcript capture, commonly referred to as dissociation bias [3]. Samples 1 and 2 also included a cryopreservation step like those employed in large-scale epidemiological studies that may have exacerbated dissociation bias."

We employed the mutual nearest number batch correction technique [1] to address potential batch effects across studies. Because of the highly cell-type specific nature of gene expression [10], we expect within person gene expression differences among cell types to be greater than the gene expression differences between people [11]. Cell type missingness or technical differences among some biological replicates may attenuate the cell type specific biological signals in gene expression patterns.

We agree that deconvolution performance could be a concern. To address these concerns, we conducted a new *in silico* deconvolution validation analysis that is now a primary figure (Figure 2). We added methods and results text for this new analysis. Our results demonstrate that our deconvolution reference is largely robust to even extremely imbalanced distributions of fetal vs. maternal cell type abundances as well as male vs. female placental cell type abundances.

Methods: **“In silico testing of deconvolution performance**

To test the performance and robustness of our placental single-cell RNA sequencing deconvolution reference, we randomly split our analytic single-cell RNA sequencing dataset into 50% training and 50% testing subsets with balanced cell type proportions [36]. We applied the test subset with CIBERSORTx Docker container (accessed 2021-12-07) to create a signature gene expression matrix to test deconvolution performance with default settings [84]. To evaluate the reference’s robustness to fetal sex and ability to discriminate immune cell types of fetal versus maternal origin, we generated *in silico* pseudo-bulk test mixtures with known distributions of fetal and maternal cells, as well as male and female placental cells. Test mixtures included all of the 50% testing data, only fetal cells from the test data, only maternal cells from the test data, only female fetal cells from the test data, or only male cells from the test data. For the female and male fetal cell test mixtures, the baseline distribution of maternal cells was maintained by randomly down-sampling the maternal cells and randomly down-sampling the male fetal cells to the number of female fetal cells. We used the signature matrix generated from the training data to estimate constituent cell type proportions in these test mixtures using CIBERSORTx with cross-platform S-mode batch correction and 50 permutations to evaluate imputation goodness-of-fit. Pearson correlations and root mean square error between the test set predicted and actual cell type proportions in the test mixtures were used to assess deconvolution performance.”

Results: **“Single-cell RNA sequencing deconvolution reference exhibits excellent in silico performance**

Based on the single cell data, we created a placental signature gene matrix that incorporated expression information across an algorithmically selected 5,229 signature genes to estimate the cellular composition of 27 fetal and maternal cell types from whole tissue gene expression data (Supplementary Figure 8). To test the performance and robustness of this placental single-cell RNA sequencing deconvolution reference, we randomly split our analytic single-cell RNA sequencing dataset into 50% training and 50% and testing subsets with balanced cell type proportions [36]. The same training dataset was used for each comparison; test mixtures were generated from the testing half of the dataset. Using a signature gene expression matrix generated from the training data, we estimated cell type composition in *in silico* pseudo-bulk testing data mixtures of known cell type composition with varying contributions of fetal vs. maternal origin cells and male vs. female fetal cells (Figure 2). In all mixtures, the 27 predicted and actual cell type proportions were correlated (p -value <0.001 for each test). In the primary deconvolution analysis of all cell types at their natural rates, estimated and actual cell type proportions had a Pearson correlation coefficient of 0.956 (95% CI [0.904, 0.980]). The worst performance was under the unrealistic scenario that the mixture was composed entirely of maternal cell types with a Pearson correlation of 0.743 (95% CI [0.505, 0.876]) between actual estimated cell type proportions. Our deconvolution reference was also robust to fetal sex when only male fetal cells (Pearson correlation = 0.894, 95% CI [0.779, 0.951]) were included, or only female fetal cells (Pearson correlation = 0.983, 95% CI [0.964, 0.993]). Together, these results show that our reference panel can successfully deconvolute placental tissues, though some maternal cell types common to both mother and fetus may be erroneously labelled fetal in the absence of fetal examples of those cell types.”

Comment 3: -There are additional published term placenta single-cell RNA-seq data - how come they were not integrated?

Response 3: This is a great point. We initially included Pique-Regi et al. 2019 [12] because of its well-documented data access via the Database of Genotypes and Phenotypes (dbGap). To address this and other comments, we have now added a third term placenta single-cell RNA-seq dataset to our paper, which increases the rigor and generalizability of our findings [2]. For resubmission, we were successful in securing data access by requesting data from the corresponding authors of Tsang et al., 2017 [2] and establishing a data use agreement. Few of the additional published studies that measure single cell gene expression in the term placenta have data readily available for reanalysis. We reached out to the corresponding authors of Zhang et al., 2021 [13] to request access to their data, but the corresponding authors were unresponsive to this request. The Zhang et al. 2021 article was published very shortly before our initial manuscript submission to *Communications Biology*.

Comment 4: -Supplementary table 3 - the main text gives p-adj but the supplementary table only has p-value. The ontology terms that are specific to a particular cell type should be visualized in some way - cell type defining processes are highlighted for SCT, but not other cell types. Do ontologies for other cell types make sense? Also for supplementary excel file it would be helpful to have ontology results for each cell type on a different tab.

Response 4: We agree there are a lot of ontology results and have revised our presentation of these findings. Due to the biological importance in placental biology of syncytiotrophoblasts, we chose to highlight their ontology results in the main manuscript text. We updated the supplementary tables (Supplementary Tables 2-3, as well as others) file to show separate cell types on each tab of the Excel file as requested that should make the information easier to parse. The p-values presented were adjusted for multiple comparison testing using the default method employed in gprofiler2's ranked gene ontology test. We updated the column headings in the supplementary table to make it clear that these p-values are adjusted for multiple comparisons and more consistent with the text.

Comment 5: -Supplementary table 4 that was downloaded seems to be different from what is referred to in the text and given in the supplementary figures file.

Response 5: Thank you for pointing out this discrepancy. The list of differentially expressed genes between proliferative cytotrophoblasts and cytotrophoblasts that was meant to be an addendum to Supplementary Figure 4 was mislabeled Supplementary Table 4. We apologize for the confusion and have made the necessary corrections. This is now Supplementary Figure 5 – addendum.

Comment 6: -It is concerning how similar the different cell types appear in figure S8 - does this indicate they were not sorted properly? One of the leukocyte samples appears to be an outlier. How would removing it impact the PCA?

Response 6: We have made several tweaks to our quality control pipeline in addressing this and other reviewer comments that are reflected throughout our updated resubmission. This figure is now more in line with expected sample clustering based on prior biological knowledge (e.g., all leukocyte samples cluster more closely with one another and the immune Hofbauer cell fractions).

Comment 7: -Presumably, cell-type defining transcripts, as determined by DESeq2 analysis, should be expressed at a higher fold in the cell type being compared to all other cell types. However, it does not appear that a fold-change cutoff was implemented according to the methods. Looking at the numbers reported in figure 2 and supplementary table 5 though, it does seem like a fold of 2 might have been used. For some cell types in table S5, the number of upregulated genes is not the same as what is depicted in figure 2.

Response 7: We agree that using a log fold-change cutoff can improve confidence in differential expression findings. We have updated the analysis to include such a cutoff and observed similar results to the initial analysis. This change is reflected in the methods section and the updated Volcano plot, which is now Supplementary Figure 11.

Revisions made:

Methods: “Cell type-defining genes were identified using the negative binomial linear model two-tailed Wald test in ‘DESeq2’ (R package, version 1.32.0) adjusted for biological replicate using default settings with contrasts comparing the expression of a gene in one cell type against the average expression across all other cell types at a false discovery rate-adjusted p-value less than 0.05 and a log-2-fold-change cutoff of 1.2 [6].”

Comment 8: -Supplementary Table S6 - how come SCT are not enriched for hormone related terms, as were enriched for the single-cell data? Similar to the single cell data, the ontology terms that are specific to a particular cell type should be visualized in some way - cell type defining processes are highlighted for SCT, but not other cell types. Also for supplementary excel file it would be helpful to have ontology results for each cell type on a different tab.

Response 8: We agree there are a lot of ontology results and have revised our presentation of these findings. Due to the biological importance in placental biology of syncytiotrophoblasts, we chose to highlight their ontology results in the main manuscript text. The syncytiotrophoblast-enriched fraction described in Supplementary Table 6 did include “Response to Estrogen” as an overrepresented biological process (false discovery-adjusted p-value < 0.001). We updated the supplementary tables (Supplementary Tables 5-6) file to show separate cell types on each tab of the Excel file as requested that should make the information easier to parse.

Comment 9: -Both Pearson and Spearman correlations are provided for single-cell vs bulk expression. Pearson correlations are very low for all cell types, and spearman is low for some of the cell types. To help convince there is a correlation, correlations between different cell types can be reported, which would presumably show that cross-cell type correlations are much lower than same cell type correlations. Are the genes considered as marker genes according to single cell data similar to those that are upregulated in the corresponding cell type in bulk data?

Response 9: We evaluated the assumptions of the parametric Pearson correlation test and determined that given the high number of zero expression values in single cell RNA sequencing results, these data do not meet the linearity assumption. The non-parametric Spearman approach is more appropriate for this data and we now provide only the Spearman coefficients and p-values.

As suggested, we also evaluated the Spearman correlations between different cell types for comparison. We added a new Supplementary Figure 13 containing the correlation coefficients between all available pairs.

Comment 10: -I am confused about the signature gene expression matrix used for deconvolution. According to figure S10, most of the genes have low expression. How would these help deconvolute data? A scalebar for expression should be provided in the figure, as well as an explanation for what is shown in the plot above the heatmap.

Response 10: We agree that the previous presentation of Supplementary Figure 10 had some limitations. A small number of very highly expressed marker genes for syncytiotrophoblast resulted in a scalebar with little dynamic range that suggests low levels of overall expression across most genes. We applied a log-transformation to handle those outliers and provide an updated supplementary figure with a larger dynamic range that includes a scalebar legend. The caption for this updated supplementary figure now also indicates that the cell types were hierarchically clustered by their signature gene expression profiles. Due to edits throughout the manuscript, this revised figure is now Supplementary Figure 8.

Comment 11: -Related to the above, no minimum gene expression cutoff is used in CIBERSORTx analysis, but wouldn't this, as well as a fold cutoff provide more robust gene signatures

Response 11: For generating the signature gene expression matrix, we used the recommended default hyperparameter settings for the CIBERSORTx program [5]. These default settings include 0 for the minimum gene expression cutoff parameter. In this case, the no minimum gene expression cutoff parameter refers to the percentage of cells in which that gene was detected, rather than the level of expression. We updated the methods text to clarify that these are the recommended default settings.

Revisions made:

Methods: " We used the CIBERSORTx Docker container (accessed 2021-12-07) to create a signature gene expression matrix for deconvolution from the counts of the single-cell RNA-sequencing data with the following default parameters: differential expression q -value <0.01 , no minimum gene expression cutoff, and a 300 gene feature selection floor and a 500 gene feature selection ceiling [5]."

Comment 12: -The authors indicate that using the cell type naive base model 1,224 genes are differentially expressed. However, this is not using a fold change, which is appropriate to use in this analysis. It looks like using fold-change would bring the number of differentially expressed genes to a very small number, making the cell type-adjusted results not too different.

Response 12: We agree that a using a log fold-change cutoff can improve confidence in differential expression findings. We have updated the analysis to include a such a cutoff and observed similar results to the initial analysis. This change is reflected in the methods section and the updated Volcano plot (Figure 4).

Revisions made:

Methods: “Differential expression analysis was conducted in limma [14] with default linear models adjusted for gestational age, fetal sex, and study source with empirical Bayes standard error moderated t-test statistics. A cell type-adjusted model was built on the base model additionally adjusted for the first five principal components of deconvoluted cell type proportions. Statistical significance was assessed at false discovery rate-adjusted q-value<0.05 and a log2 fold change cutoff of 0.1.”

Comment 13: -It appears that even if a fold-change cutoff were to be used, FLT1 and FSTL3, both which have been associated PE (the authors do note this and do further analysis with FLT1), would have been identified in the standard differential expression analysis, but not in the model that considers cell type. Does using deconvolution then lead to us losing information about differentially expressed genes in PE? Or is the point to use both methods to look at genes and then possible cell types that are impacted?

Response 13: This is an excellent point. We believe our results demonstrate the importance of considering cell type proportions in analysis of bulk tissue transcriptomic investigations. Delineating whether a phenotype such as preeclampsia is associated with differential cell type proportions versus intracellular changes to gene expression have important implications for understanding the etiology of the phenotype under investigation. Our results suggest that, in this case, differential cell type proportions by preeclampsia case-control status are a predictor of differential *FLT1* expression results. Incomplete mediation (we observed ~37.8% mediation) suggests that there are also intracellular changes to *FLT1* gene expression. We have made some modifications to the discussion to better communicate this.

Revisions made:

Discussion: “These results suggest that placental cell type proportion differences may be an overlooked factor in explaining the well-documented association between preeclampsia and *FLT1*, *LEP*, and *ENG* expression [15–18]. Downregulation of mitochondrial biogenesis, aerobic respiration, and ribosome biogenesis were robust to cell type adjustment, suggesting intracellular changes to these pathways.”

Discussion: “Future work should consider and account for cell type proportions and the cell type-specific expression patterns of genes that regulate placental development or are associated with preeclampsia to better understand preeclampsia etiology.”

Comment 14: -It would be helpful to know what numbers were used for filtering criteria in scRNAseq (rather than just exceeding four median absolute deviations). How come 4 deviations were used for samples 1 and 2 and 3 median deviations were used for samples 3-5?

Response 14: Yes, transparency in quality control metrics is very important. With the additional Tsang Samples we added, we reperformed quality control. Per current recommendations [19], each batch should be quality-controlled separately. We used a fixed threshold for number of RNA molecules and genes detected to avoid including cells with excessively low counts on these metrics. Because Michigan Samples 1-2 had a somewhat higher distribution quality-control metrics on average due to the relatively higher proportion of peripheral maternal immune cells that are less susceptible to dissociation bias, we expanded the adaptive threshold for mitochondrial gene expression to be more comparable to the other datasets. We have updated the text to reflect these details, including providing the absolute value of these thresholds.

Revisions made:

Methods: “Per cell quality control criteria were calculated using the quickQCPerCell() function (scater R package, version 1.18.6) with default settings [20] (Supplementary Figures 1-2) and included total RNA molecules, unique genes, and percentage of reads mapping to mitochondrial genes [21]. According to current recommended best practice, each batch was quality-controlled separately [19]. We excluded 6,497 low-quality outlier cells defined as cells with less than 500 unique RNA molecules, less than 200 unique genes, or that were outliers in mitochondrial gene mapping rate. Mitochondrial mapping outliers exceeded four median absolute deviations in samples 1 and 2 (mitochondrial reads > 9.2%) or three median absolute deviations in samples 3, 4, and 5 (mitochondrial reads > 8.9%) and samples 6, 7, 8C, 8P, 9C, and 9P (mitochondrial reads > 9.1%).”

Comment 15: -Should male and female be separated for the purpose of deconvolution? There are 2 biological female and 3 biological male studies.

Response 15: We agree that this is a relevant point of concern. To address these concerns, we conducted an *in silico* deconvolution validation analysis that is now a primary figure (Figure 2). We added methods and results text for this new analysis. Our results demonstrate that our deconvolution reference is largely robust to even extremely imbalanced distributions of male vs. female placental cell type abundances.

Revisions made:

Methods: “**In silico testing of deconvolution performance**

To test the performance and robustness of our placental single-cell RNA sequencing deconvolution reference, we randomly split our analytic single-cell RNA sequencing dataset into 50% training and 50% testing subsets with balanced cell type proportions [36]. We applied the test subset with CIBERSORTx Docker container (accessed 2021-12-07) to create a signature gene expression matrix to test deconvolution performance with default settings [84]. To evaluate the reference’s robustness to fetal sex and ability to discriminate immune cell types of fetal versus maternal origin, we generated *in silico* pseudo-bulk test mixtures with known distributions of fetal and maternal cells, as well as male and female placental cells. Test mixtures included all of the 50% testing data, only fetal cells from the test data, only maternal cells from the test data, only female fetal cells from the test data, or only male cells from the test data. For the female and male fetal cell test mixtures, the baseline distribution of maternal cells was maintained by randomly down-sampling the maternal cells and randomly down-sampling the male fetal cells to the number of female fetal cells. We used the signature matrix generated from the training data to estimate constituent cell type proportions in these test mixtures using CIBERSORTx with cross-platform S-mode batch correction and 50 permutations to evaluate imputation goodness-of-fit. Pearson correlations and root mean square error between the test set predicted and actual cell type proportions in the test mixtures were used to assess deconvolution performance.”

Results: “**Single-cell RNA sequencing deconvolution reference exhibits excellent in silico performance**

Based on the single cell data, we created a placental signature gene matrix that incorporated expression information across an algorithmically selected 5,229 signature genes to estimate the cellular composition of 27 fetal and maternal cell types from whole tissue gene expression data (Supplementary Figure 8). To test the performance and robustness of this placental single-cell RNA sequencing deconvolution reference, we randomly split our analytic single-cell RNA sequencing

dataset into 50% training and 50% and testing subsets with balanced cell type proportions [36]. The same training dataset was used for each comparison; test mixtures were generated from the testing half of the dataset. Using a signature gene expression matrix generated from the training data, we estimated cell type composition in in silico pseudo-bulk testing data mixtures of known cell type composition with varying contributions of fetal vs. maternal origin cells and male vs. female fetal cells (Figure 2). In all mixtures, the 27 predicted and actual cell type proportions were correlated (p -value <0.001 for each test). In the primary deconvolution analysis of all cell types at their natural rates, estimated and actual cell type proportions had a Pearson correlation coefficient of 0.956 (95% CI [0.904, 0.980]). The worst performance was under the unrealistic scenario that the mixture was composed entirely of maternal cell types with a Pearson correlation of 0.743 (95% CI [0.505, 0.876]) between actual estimated cell type proportions. Our deconvolution reference was also robust to fetal sex when only male fetal cells (Pearson correlation = 0.894, 95% CI [0.779, 0.951]) were included, or only female fetal cells (Pearson correlation = 0.983, 95% CI [0.964, 0.993]). Together, these results show that our reference panel can successfully deconvolute placental tissues, though some maternal cell types common to both mother and fetus may be erroneously labelled fetal in the absence of fetal examples of those cell types.”

Comment 16: -Supplementary table 2 would be more useful if there was a separate tab for each cell type, and each tab had only the significantly upregulated markers for that cell type.

Response 16: We agree that Supplementary Table 2 would be more useful with these recommended changes and have made those changes.

Comment 17: -Supplementary table 5 - it would be helpful if results for each cell type was in a different tab

Response 17: We have revised the table as suggested.

Comment 18: -The authors provide a github page for the lab, but there is no specific information about this study on that page. Also, the specific link to this study should be provided in the paper, rather than the lab page.

Response 18: We now provide a GitHub link that directly accesses the project for this paper (<https://github.com/bakulskilab/Placental-gene-expression-based-cell-type-deconvolution-Cell-proportions-drive-preeclampsia-gene-exp>).

References Cited

1. Haghverdi L, Lun ATL, Morgan MD, Marioni JC (2018) Batch effects in single-cell RNA-sequencing data are corrected by matching mutual nearest neighbors. *Nature Biotechnology* 36:421–427. <https://doi.org/10.1038/nbt.4091>
2. Tsang JCH, Vong JSL, Ji L, et al (2017) Integrative single-cell and cell-free plasma RNA transcriptomics elucidates placental cellular dynamics. *PNAS* 114:E7786–E7795. <https://doi.org/10.1073/pnas.1710470114>

3. Hedlund E, Deng Q (2018) Single-cell RNA sequencing: Technical advancements and biological applications. *Molecular Aspects of Medicine* 59:36–46. <https://doi.org/10.1016/j.mam.2017.07.003>
4. Denisenko E, Guo BB, Jones M, et al (2020) Systematic assessment of tissue dissociation and storage biases in single-cell and single-nucleus RNA-seq workflows. *Genome Biol* 21:130. <https://doi.org/10.1186/s13059-020-02048-6>
5. Steen CB, Liu CL, Alizadeh AA, Newman AM (2020) Profiling Cell Type Abundance and Expression in Bulk Tissues with CIBERSORTx. *Methods Mol Biol* 2117:135–157. https://doi.org/10.1007/978-1-0716-0301-7_7
6. Love MI, Huber W, Anders S (2014) Moderated estimation of fold change and dispersion for RNA-seq data with DESeq2. *Genome Biol* 15:550. <https://doi.org/10.1186/s13059-014-0550-8>
7. Yabe S, Alexenko AP, Amita M, et al (2016) Comparison of syncytiotrophoblast generated from human embryonic stem cells and from term placentas. *PNAS* 113:E2598–E2607. <https://doi.org/10.1073/pnas.1601630113>
8. Li L, Schust DJ (2015) Isolation, purification and in vitro differentiation of cytotrophoblast cells from human term placenta. *Reprod Biol Endocrinol* 13:. <https://doi.org/10.1186/s12958-015-0070-8>
9. Petroff MG, Phillips TA, Ka H, et al (2006) Isolation and culture of term human trophoblast cells. *Methods Mol Med* 121:203–217
10. Newman AM, Steen CB, Liu CL, et al (2019) Determining cell type abundance and expression from bulk tissues with digital cytometry. *Nature Biotechnology* 37:773–782. <https://doi.org/10.1038/s41587-019-0114-2>
11. Avila Cobos F, Alquicira-Hernandez J, Powell JE, et al (2020) Benchmarking of cell type deconvolution pipelines for transcriptomics data. *Nat Commun* 11:5650. <https://doi.org/10.1038/s41467-020-19015-1>
12. Pique-Regi R, Romero R, Tarca AL, et al (2019) Single cell transcriptional signatures of the human placenta in term and preterm parturition. *eLife* 8:. <https://doi.org/10.7554/eLife.52004>
13. Zhang T, Bian Q, Chen Y, et al (2021) Dissecting human trophoblast cell transcriptional heterogeneity in preeclampsia using single-cell RNA sequencing. *Mol Genet Genomic Med* e1730. <https://doi.org/10.1002/mgg3.1730>
14. Smyth GK (2005) limma: Linear Models for Microarray Data. In: Gentleman R, Carey VJ, Huber W, et al (eds) *Bioinformatics and Computational Biology Solutions Using R and Bioconductor*. Springer New York, New York, NY, pp 397–420
15. Enquobahrie DA, Meller M, Rice K, et al (2008) Differential placental gene expression in preeclampsia. *Am J Obstet Gynecol* 199:566.e1-566.11. <https://doi.org/10.1016/j.ajog.2008.04.020>
16. Várkonyi T, Nagy B, Füle T, et al (2011) Microarray Profiling Reveals That Placental Transcriptomes of Early-onset HELLP Syndrome and Preeclampsia Are Similar. *Placenta* 32:S21–S29. <https://doi.org/10.1016/j.placenta.2010.04.014>
17. Vennou KE, Kontou PI, Braliou GG, Bagos PG (2020) Meta-analysis of gene expression profiles in preeclampsia. *Pregnancy Hypertension* 19:52–60. <https://doi.org/10.1016/j.preghy.2019.12.007>
18. Sitras V, Fenton C, Acharya G (2015) Gene expression profile in cardiovascular disease and preeclampsia: A meta-analysis of the transcriptome based on raw data from human studies deposited in Gene Expression Omnibus. *Placenta* 36:170–178. <https://doi.org/10.1016/j.placenta.2014.11.017>

19. Chapter 1 Correcting batch effects | Multi-Sample Single-Cell Analyses with Bioconductor
20. Scater: pre-processing, quality control, normalization and visualization of single-cell RNA-seq data in R | Bioinformatics | Oxford Academic. <https://academic.oup.com/bioinformatics/article/33/8/1179/2907823>. Accessed 28 Jan 2021
21. Luecken MD, Theis FJ (2019) Current best practices in single-cell RNA-seq analysis: a tutorial. *Molecular Systems Biology* 15:e8746. <https://doi.org/10.15252/msb.20188746>

Reviewers' comments:

Reviewer #1 (Remarks to the Author):

The manuscript by Campbell et al. entitled "Placental gene expression-based cell type deconvolution: Cell proportions drive preeclampsia gene expression differences" explores using a reference-based deconvolution approach for placental gene-expression using data derived from single-cell RNAseq (scRNAseq). The authors applied the approach to 330 publicly available samples to test associations with preeclampsia. The authors identified 19 fetal and eight maternal cell types in their reference and found that no genes were differentially expressed after correcting for cell types. The idea is enticing, and this work is trying to fill a critical gap in reproductive epidemiology. The authors have added a new dataset for their references, and solved several outstanding problems.

Comment: Figure 1: the labels for CD8 T cytotoxic are still the same instead of "activated"

Supplementary Tables: I am having a hard time trying to figure out the Excel files. They do not have a heading or any indicator of which table is which. Currently, I cannot provide a lot of feedback on that.

Supplementary table 7: I am confused with the results, it seems that the cells are not matching the signatures. This needs some explanation. For example, endothelial is mostly cytotrophoblast.

Reviewer #2 (Remarks to the Author):

In the revised manuscript, the authors have addressed most of the previous concerns. Overall, the manuscript is clearly written and presents novel analysis methods and very interesting results that are important for the research community. I have minor comments (most relating to my previous comments), that I think can be easily addressed now that the more substantial revisions are complete. I think addressing these comments will strengthen/tighten the manuscript.

-I'm a little confused about the different presentation of functional enrichment analysis in supplementary table 3 and supplementary table 6. Were different tools used for the analysis? If so, why? The data in supplementary table 6 seems noisier, and has a lot of terms, even related to cell migration enriched in syncytiotrophoblasts. Given this, how were the terms that were presented in the text decided? One term highlighted as cell type-defining for syncytiotrophoblasts is 'stem cell differentiation', but similar terms are also enriched in extravillous trophoblasts. In that case, should it be listed for syncytiotrophoblasts?

-The authors refer to loosely defined clusters in PCA plots shown in supplementary figure 10. It is difficult to see how the conclusions were made. To my eye, the cell types are not clustering as they should. Could the authors provide more quantitative analysis to determining how the clusters are defined? Without this, it seems the expected cell types are not grouping together.

-The authors show a low correlation between expression of genes in sc data and sorted data. As they mention, this is likely due to comparison of different types of data. Therefore, the more relevant comparison seems to be if the cell-type specific markers determined using each method have significant overlap.

-The authors describe the lack of certain trophoblast in scRNAseq due to dissociation bias. It is possible that snRNA-seq would allow sequencing of syncytiotrophoblasts which are more fragile in nature, as described here: <https://www.ncbi.nlm.nih.gov/pmc/articles/PMC8334858/>. Of course I am not suggesting the authors carry out snRNAseq, just acknowledging the issue further in the discussion might direct other researchers to using snRNAseq when working with syncytiotrophoblasts.

-As the authors mention, the manuscript would be strengthened by incorporation of existing preeclamptic placenta data (e.g. <https://pubmed.ncbi.nlm.nih.gov/35289305/>). However, the data/analysis presented is excellent on its own as well. Perhaps they could cite this or similar papers when writing about this point in the discussion.

-The methods indicate that the freemuxlet program/popsicle package were used to assign fetal or maternal origin. A quick look at the package, it is not too clear how it is used for assigning sex.

-From methods - cell types were assigned based on canonical cell type marker gene expression and marker genes - it would be helpful to have the cell types/marker genes used as a supplementary table.

We thank the editor for the opportunity to resubmit this manuscript. While the reviewers highlighted the importance of this research, they identified several areas for improvement of the research. In response to the reviewer feedback, we updated the manuscript by aligning our gene ontology enrichment testing methods for the sorted cell type data with the single-cell testing approach, updating the sorted vs. single-cell cell type comparison approach to the differentially expressed gene and associated pathway level, and providing a marker gene expression matrix figure to more clearly communicate how we annotated single-cell cell type clusters. Our specific changes are described point by point below.

Reviewers' comments:

Reviewer #1 (Remarks to the Author):

Comment 1: The manuscript by Campbell et al. entitled "Placental gene expression-based cell type deconvolution: Cell proportions drive preeclampsia gene expression differences" explores using a reference-based deconvolution approach for placental gene-expression using data derived from single-cell RNAseq (scRNAseq). The authors applied the approach to 330 publicly available samples to test associations with preeclampsia. The authors identified 19 fetal and eight maternal cell types in their reference and found that no genes were differentially expressed after correcting for cell types. The idea is enticing, and this work is trying to fill a critical gap in reproductive epidemiology. The authors have added a new dataset for their references, and solved several outstanding problems.

Response 1: We thank the reviewer for recognizing the utility of this research for reproductive epidemiology and continuing to provide helpful feedback.

Comment 2: Figure 1: the labels for CD8 T cytotoxic are still the same instead of "activated"

Response 2: We corrected this labelling error in Figure 1.

Comment 3: Supplementary Tables: I am having a hard time trying to figure out the Excel files. They do not have a heading or any indicator or which table is which. Currently, I cannot provide a lot of feedback on that.

Response 3: We understand that there are a lot of supplemental tables that contain a lot of information. We provided detailed table descriptions in the supplementary document. We updated Excel file tabs with clearer descriptions.

Comment 4: Supplementary table 7: I am confused with the results, it seems that the cells are not matching the signatures. This needs some explanation. For example, endothelial is mostly cytotrophoblast.

Response 4: FACS-sorted purity estimates for some cell types were low contributing to this mismatch. We expected challenges with FACS in term placental tissue due to apoptosis

characteristic of placental development and parturition, potentially leading to degradation of cell surface markers or cell integrity. Thus, our primary analyses focused on the single cell RNA sequencing, which was not dependent on cell surface markers. We have addressed this disparity in the the discussion.

Discussion section: “Similarly, the sample size of FACS-sorted tissues was limited, and some cell type fractions were excluded due to low RNA quality or exhibited poor estimated purity, likely complicated by degradation of cell surface markers from apoptosis characteristic of development and parturition [1, 2] and sample processing.”

Reviewer #2 (Remarks to the Author):

Comment 1: In the revised manuscript, the authors have addressed most of the previous concerns. Overall, the manuscript is clearly written and presents novel analysis methods and very interesting results that are important for the research community. I have minor comments (most relating to my previous comments), that I think can be easily addressed now that the more substantial revisions are complete. I think addressing these comments will strengthen/tighten the manuscript.

Response 1: We thank the reviewer for their nuanced and thoughtful feedback to improve the manuscript.

Comment 2: I'm a little confused about the different presentation of functional enrichment analysis in supplementary table 3 and supplementary table 6. Were different tools used for the analysis? If so, why? The data in supplementary table 6 seems noisier, and has a lot of terms, even related to cell migration enriched in syncytiotrophoblasts. Given this, how were the terms that were presented in the text decided? One term highlighted as cell type-defining for syncytiotrophoblasts is 'stem cell differentiation', but similar terms are also enriched in extravillous trophoblasts. In that case, should it be listed for syncytiotrophoblasts?

Response 2: We agree that comparisons between single- and sorted-cell type data would be facilitated by using common analytic approaches. To make the results more comparable, we now updated the sorted cell type gene ontology enrichment results to the same gProfiler2 test as we used with the single-cell data. The new sorted cell type gene ontology enrichment results are available in **Supplementary Table 6**.

Because we used a one vs. all differential expression approach to identify overexpressed genes by cell type, we would expect to see an overlap in overrepresented pathways between related cell types. While all enriched ontologies are available, we chose to highlight terms in the text that are biologically relevant to placental structure and function beyond pathways that are shared broadly across lineages. We have clarified these points in the results section for both differential expression analyses.

Results:

“ To identify upregulated genes in each cell type, we compared the expression of a gene in one cell type against that gene’s average expression in all other cell types (**Supplementary Table 2**). Consequently, the same genes could be upregulated across several cell types of similar lineage. *FLT1* expression was highly upregulated in extravillous trophoblasts (\log_2 fold change (FC)=3.89,

$p_{\text{adj}} < 0.001$). Trophoblast cell types had the largest and most diverse transcriptomes, characterized by the largest number of unique RNA transcripts and detected genes per cell (**Supplementary Figure 7**). Functional analysis of upregulated genes revealed cell type biological processes (**Supplementary Table 3**). For example, fetal extravillous trophoblasts were enriched for genes relevant to placental structure and function such as cell migration ($p_{\text{adj}} < 0.001$) and response to oxygen levels ($p_{\text{adj}} < 0.001$) and syncytiotrophoblasts were enriched for genes involved in steroid hormone biosynthetic process ($p_{\text{adj}} < 0.001$). Technical replication in Michigan samples 1 and 2 appeared high in UMAP space (**Supplementary Figure 8A-B**). Indeed, the average intra-cluster gene expression between technical replicates had an average Spearman correlation (mean \pm standard deviation) of 0.94 ± 0.14 for sample 1 and 0.88 ± 0.20 for sample 2 (p -values < 0.001)."

" To identify upregulated genes in each cell type, we compared the expression of a gene in one cell type against that gene's average expression in all other cell types (**Supplementary Figure 12**). Consequently, the same genes could be upregulated across several cell types of similar lineage. All 38,468 uniquely mapping genes were tested. 746 genes were algorithmically dropped from the syncytiotrophoblast contrast due to excessively low counts, low variability, or extreme outlier status. Large-scale gene expression differences were observed for each cell type (**Supplementary Table 5**). Functional analysis of upregulated genes revealed cell type biological processes (**Supplementary Table 6**). For example, syncytiotrophoblasts were enriched for genes relevant to placental structure and function such as angiogenesis, cell-substrate adhesion, and regulation of epithelial cell proliferation ($p_{\text{adj}} < 0.001$). To compare sorted and single-cell differential expression and enrichment results, we tabulated the number of unique genes and pathways overlapping between the two analyses after collapsing the single-cell cell type cluster labels to the seven cell type fractions we targeted for isolation for downstream analyses (**Supplementary Table 7**). On average, 15.0% of single-cell upregulated genes and 5.9% of enriched pathways were also identified among the sorted cell data. On average, 17.5% of sorted cell type upregulated genes and 39.2% of pathways were also identified among the single-cell data. Sorted endothelial cells results were limited due to the limited number of biological replicates."

Comment 3: The authors refer to loosely defined clusters in PCA plots shown in supplementary figure 10. It is difficult to see how the conclusions were made. To my eye, the cell types are not clustering as they should. Could the authors provide more quantitative analysis to determining how the clusters are defined? Without this, it seems the expected cell types are not grouping together.

Response 3: Thank you for pointing this out. We did not perform clustering analysis with the sorted cell type data. Consequently, we adjusted the text to reflect that the PCA plot in **Supplementary Figure 11** is only meant to provide a high-level view of the sorted cell type gene expression data.

Methods: "Principal components analysis of whole-transcriptome sorted-cell bulk RNA sequencing normalized counts is also provided (**Supplementary Figure 11**)."

Comment 4: The authors show a low correlation between expression of genes in sc data and sorted data. As they mention, this is likely due to comparison of different types of data. Therefore, the more relevant comparison seems to be if the cell-type specific markers determined using each method have significant overlap.

Response 4: We agree that a more apt comparison between single- and sorted-cell type data would be performed using the results appropriate to each platform, e.g. differentially expressed genes and enriched pathways among those genes. We removed the previous analysis of the correlation between sorted and single-cell gene expression values. In its place we tabulated the number of upregulated genes and enriched pathways for each analysis and cell type as well as the number of unique overlapping genes and pathways in **Supplementary Table 7**. These results have been added to the results section with some additional clarification about the differential gene expression test.

Results:

“ To identify upregulated genes in each cell type, we compared the expression of a gene in one cell type against that gene’s average expression in all other cell types (**Supplementary Figure 12**). Consequently, the same genes could be upregulated across several cell types of similar lineage. All 38,468 uniquely mapping genes were tested. 746 genes were algorithmically dropped from the syncytiotrophoblast contrast due to excessively low counts, low variability, or extreme outlier status. Large-scale gene expression differences were observed for each cell type (**Supplementary Table 5**). Functional analysis of upregulated genes revealed cell type biological processes (**Supplementary Table 6**). For example, syncytiotrophoblasts were enriched for genes relevant to placental structure and function such as angiogenesis, cell-substrate adhesion, and regulation of epithelial cell proliferation ($p_{adj}<0.001$). To compare sorted and single-cell differential expression and enrichment results, we tabulated the number of unique genes and pathways overlapping between the two analyses after collapsing the single-cell cell type cluster labels to the seven cell type fractions we targeted for isolation for downstream analyses (**Supplementary Table 7**). On average, 15.0% of single-cell upregulated genes and 5.9% of enriched pathways were also identified among the sorted cell data. On average, 17.5% of sorted cell type upregulated genes and 39.2% of pathways were also identified among the single-cell data. Sorted endothelial cells results were limited due to the limited number of biological replicates.”

Comment 5: The authors describe the lack of certain trophoblast in scRNAseq due to dissociation bias. It is possible that snRNA-seq would allow sequencing of syncytiotrophoblasts which are more fragile in nature, as described here: <https://www.ncbi.nlm.nih.gov/pmc/articles/PMC8334858/> . Of course I am not suggesting the authors carry out snRNAseq, just acknowledging the issue further in the discussion might direct other researchers to using snRNAseq when working with syncytiotrophoblasts.

Response 5: We thank the reviewer for this insightful comment. We agree that acknowledging this potential alternative to readers is of benefit. We updated the discussion section to reflect this.

Discussion: “Future studies may propose alternative approaches to perform unbiased single-cell RNA sequencing in placental tissues; indeed, single nucleus RNA sequencing has been used to characterize syncytiotrophoblast and may be more appropriate to assay such cell types sensitive to dissociation procedures [3].”

Comment 6: As the authors mention, the manuscript would be strengthened by incorporation of existing preeclamptic placenta data (e.g. <https://pubmed.ncbi.nlm.nih.gov/35289305/>). However, the data/analysis presented is excellent on its own as well. Perhaps they could cite this or similar papers

when writing about this point in the discussion.

Response 6: We thank the reviewer for their expert insight on this topic. We have added this citation to our discussion section because it provides relevant context to our discussion of extravillous trophoblast function in preeclampsia.

Discussion: “A recent single-cell RNA-sequencing case-control study of preeclampsia, however, identified upregulation of TGFB1 in extravillous trophoblasts, potentially indicative of altered trophoblast differentiation or invasion [80, 81]. A similar study revealed decreased activity of gene network modules regulated by transcription factors ATF3, CEBPB, and GTF2B and decreased expression of CEBPB and GTF2B in preeclamptic extravillous trophoblasts compared to controls; follow-up in vitro experiments suggested CEBPB and GTF2B knockdown reduced extravillous trophoblast viability and invasion [4].”

Comment 7: The methods indicate that the freemuxlet program/popsicle package were used to assign fetal or maternal origin. A quick look at the package, it is not too clear how it is used for assigning sex.

Response 7: We updated the methods section to make this clearer. Yes, we used the freemuxlet program/popsicle package to assign fetal or maternal origin. However, sex assignment was determined by XIST expression. These data are summarized in **Supplementary Figure 4**.

Methods: “Fetal sex in Michigan (this study) samples was determined with average normalized XIST expression; fetal sex in Pique-Regi and Tsang samples was determined by annotation and confirmed with average normalized XIST expression (**Supplementary Figure 4**).

Comment 8: From methods - cell types were assigned based on canonical cell type marker gene expression and marker genes - it would be helpful to have the cell types/marker genes used as a supplementary table.

Response 8: We agree that this would be helpful information for the manuscript. We have included marker gene information to the results section describing how cell type labels were annotated to cell clusters. We have additionally added a dot plot to show how marker gene expression maps onto cell types.

Results:

“ Cell type clustering decisions balanced cluster stability, resolution, and biologic plausibility with prior knowledge. If desired, downstream analyses could collapse cell subtypes into a single, more general cell type cluster. We observed placenta-specific trophoblast cell types including cytotrophoblasts (*KRT7*), proliferative cytotrophoblasts (*KRT7*, *STMN1* and other proliferation-related genes) [5], extravillous trophoblasts (*HLA-G*) [6], and syncytiotrophoblasts (*PSG4* and other pregnancy-specific hormone genes) (**Supplementary Figure 5A**) [7]. Proliferative cytotrophoblasts were distinguished from other cytotrophoblasts by overexpression of genes related to cytoplasmic translation ($p_{\text{adj}}=8.1 \times 10^{-15}$) and mitotic sister chromatin segregation ($p_{\text{adj}}=1.5 \times 10^{-12}$), indicating their proliferative phenotype (**Supplementary Figure 6**). Other fetal-specific cell types included mesenchymal stem cells (*COL1A1^{lo}*, *TAGLN^{lo}*, *LUM^{hi}*), fibroblasts (*COL1A1^{hi}*, *TAGLN^{hi}*, *LUM^{lo}*) [8], endothelial cells (*PECAM1*) [9], and Hofbauer cells (*CD163*) [10] (**Figure 1B**).

Fetal and maternal lymphocytes, B cells, and monocytes were also captured (**Figure 1B-C**). We observed fetal and maternal B cells (*CD79A*) [11], maternal plasma cells (*XBP1*, *IGHA* and other immunoglobulins) [12]. We also observed fetal and maternal CD14⁺ monocytes (*CD14⁺/FCGR3A⁻*), maternal CD16⁺ monocytes (*CD14⁺/FCGR3A⁺*) [13], and a small population of fetal plasmacytoid dendritic-like cells (*FLT3⁺/ITM2C⁺*) [14, 15]. We further observed fetal and maternal natural killer cells (*NKG7*) and fetal GZMB⁺ or GZMK⁺ natural killer cell subtypes, and fetal natural killer T cells (*NKG7⁺/CD3E⁺/CD8A⁻*) [16, 17]. Finally, we observed a variety of T cell subtypes: naïve CD4⁺ (*CCR7*, *CD3E*, *CD4*), naïve CD8⁺ (*CCR7*, *CD3E*, *CD8A*), memory CD4⁺ (*S100A4*, *CD3E*, *CD4*, *IL2*, *CCR7^{lo}*), and activated CD8⁺ T cells (*NKG7*, *CD3E*, *CD8A*) (**Supplementary Figure 5B**) [18].”

References Cited

1. Phillippe M (2015) Cell-Free Fetal DNA, Telomeres, and the Spontaneous Onset of Parturition. *Reprod Sci* 22:1186–1201. <https://doi.org/10.1177/1933719115592714>
2. Sharp AN, Heazell AEP, Crocker IP, Mor G (2010) Placental Apoptosis in Health and Disease. *American Journal of Reproductive Immunology* 64:159–169. <https://doi.org/10.1111/j.1600-0897.2010.00837.x>
3. Khan T, Seetharam AS, Zhou J, et al (2021) Single Nucleus RNA Sequence (snRNAseq) Analysis of the Spectrum of Trophoblast Lineages Generated From Human Pluripotent Stem Cells in vitro. *Front Cell Dev Biol* 9:695248. <https://doi.org/10.3389/fcell.2021.695248>
4. Zhou W, Wang H, Yang Y, et al (2022) Trophoblast Cell Subtypes and Dysfunction in the Placenta of Individuals with Preeclampsia Revealed by Single-Cell RNA Sequencing. *Mol Cells* 45:317–328. <https://doi.org/10.14348/molcells.2021.0211>
5. Bulmer JN, Morrison L, Johnson PM (1988) Expression of the proliferation markers Ki67 and transferrin receptor by human trophoblast populations. *J Reprod Immunol* 14:291–302. [https://doi.org/10.1016/0165-0378\(88\)90028-9](https://doi.org/10.1016/0165-0378(88)90028-9)
6. Gonen-Gross T, Goldman-Wohl D, Huppertz B, et al (2010) Inhibitory NK Receptor Recognition of HLA-G: Regulation by Contact Residues and by Cell Specific Expression at the Fetal-Maternal Interface. *PLoS One* 5:. <https://doi.org/10.1371/journal.pone.0008941>
7. Zhou GQ, Baranov V, Zimmermann W, et al (1997) Highly specific monoclonal antibody demonstrates that pregnancy-specific glycoprotein (PSG) is limited to syncytiotrophoblast in human early and term placenta. *Placenta* 18:491–501. [https://doi.org/10.1016/0143-4004\(77\)90002-9](https://doi.org/10.1016/0143-4004(77)90002-9)
8. Baboolal TG, Boxall SA, Churchman SM, et al (2014) Intrinsic multipotential mesenchymal stromal cell activity in gelatinous Heberden’s nodes in osteoarthritis at clinical presentation. *Arthritis Res Ther* 16:R119. <https://doi.org/10.1186/ar4574>
9. Dejana E (2004) Endothelial cell–cell junctions: happy together. *Nat Rev Mol Cell Biol* 5:261–270. <https://doi.org/10.1038/nrm1357>
10. Tang Z, Tadesse S, Norwitz E, et al (2011) Isolation of Hofbauer Cells from Human Term Placentas with High Yield and Purity. *American Journal of Reproductive Immunology* 66:336–348. <https://doi.org/10.1111/j.1600-0897.2011.01006.x>
11. van Noesel CJ, van Lier RA, Cordell JL, et al (1991) The membrane IgM-associated heterodimer on human B cells is a newly defined B cell antigen that contains the protein product of the mb-1 gene. *J Immunol* 146:3881–3888

12. Shapiro-Shelef M, Calame K (2005) Regulation of plasma-cell development. *Nat Rev Immunol* 5:230–242. <https://doi.org/10.1038/nri1572>
13. Geissmann F, Jung S, Littman DR (2003) Blood Monocytes Consist of Two Principal Subsets with Distinct Migratory Properties. *Immunity* 19:71–82. [https://doi.org/10.1016/S1074-7613\(03\)00174-2](https://doi.org/10.1016/S1074-7613(03)00174-2)
14. Villani A-C, Satija R, Reynolds G, et al (2017) Single-cell RNA-seq reveals new types of human blood dendritic cells, monocytes, and progenitors. *Science* 356:.. <https://doi.org/10.1126/science.aah4573>
15. Musumeci A, Lutz K, Winheim E, Krug AB (2019) What Makes a pDC: Recent Advances in Understanding Plasmacytoid DC Development and Heterogeneity. *Frontiers in Immunology* 10:
16. Zhao Y, Li X, Zhao W, et al (2019) Single-cell transcriptomic landscape of nucleated cells in umbilical cord blood. *Gigascience* 8:giz047. <https://doi.org/10.1093/gigascience/giz047>
17. Yang C, Siebert JR, Burns R, et al (2019) Heterogeneity of human bone marrow and blood natural killer cells defined by single-cell transcriptome. *Nat Commun* 10:3931. <https://doi.org/10.1038/s41467-019-11947-7>
18. Santana MA, Esquivel-Guadarrama F (2006) Cell Biology of T Cell Activation and Differentiation. In: *International Review of Cytology*. Academic Press, pp 217–274

REVIEWERS' COMMENTS:

Reviewer #1 (Remarks to the Author):

The authors have addressed all my comments. I have no additional comments.

Reviewer #2 (Remarks to the Author):

The authors have addressed my comments and I believe this is an excellent manuscript.